# Single cell transcriptomic analysis reveals cellular diversity of murine esophageal epithelium

Mohammad Faujul Kabir[1,8], Adam L. Karami[1,8], Ricardo Cruz-Acuña[2], Alena Klochkova[1], Reshu Saxena[1], Anbin Mu[1], Mary Grace Murray [1], Jasmine Cruz[1], Annie D. Fuller[1], Margarette H. Clevenger[3], Kumaraswamy Naidu Chitrala[1], Yinfei Tan[4], Kelsey Keith [5], Jozef Madzo [5], Hugh Huang[5], Jaroslav Jelinek [5], Tatiana Karakasheva [6], Kathryn E. Hamilton[6], Amanda B. Muir[6], Marie-Pier Tétreault[3] & Kelly A. Whelan [1,7✉]

Although morphologic progression coupled with expression of specific molecular markers has been characterized along the esophageal squamous differentiation gradient, the molecular heterogeneity within cell types along this trajectory has yet to be classified at the single cell level. To address this knowledge gap, we perform single cell RNA-sequencing of 44,679 murine esophageal epithelial, to identify 11 distinct cell populations as well as pathways alterations along the basal-superficial axis and in each individual population. We evaluate the impact of aging upon esophageal epithelial cell populations and demonstrate age-associated mitochondrial dysfunction. We compare single cell transcriptomic profiles in 3D murine organoids and human esophageal biopsies with that of murine esophageal epithelium. Finally, we employ pseudotemporal trajectory analysis to develop a working model of cell fate determination in murine esophageal epithelium. These studies provide comprehensive molecular perspective on the cellular heterogeneity of murine esophageal epithelium in the context of homeostasis and aging.

[1] Fels Cancer Institute for Personalized Medicine, Temple University Lewis Katz School of Medicine, Philadelphia, PA, USA. [2] Division of Digestive and Liver Diseases, Department of Medicine, Columbia University Medical Center, New York, NY, USA. [3] Department of Medicine, Gastroenterology and Hepatology Division, Northwestern University Feinberg School of Medicine, Chicago, IL, USA. [4] Fox Chase Cancer Center, Philadelphia, PA, USA. [5] Coriell Institute for Medical Research, Camden, NJ, USA. [6] Department of Pediatrics, Division of Gastroenterology, Hepatology, and Nutrition, Children's Hospital of Philadelphia, Philadelphia, PA, USA. [7] Department of Cancer & Cellular Biology, Temple University Lewis Katz School of Medicine, Philadelphia, PA, USA. [8] These authors contributed equally: Mohammad Faujul Kabir, Adam L. Karami. ✉email: kelly.whelan@temple.edu

In stratified squamous epithelium of the esophagus, basal cells give rise to overlying keratinocytes that exhibit a gradient of squamous differentiation as they move toward the lumen and ultimately desquamate. Squamous cell differentiation (SCD) in esophageal keratinocytes is marked by downregulation of basal cell markers, including cytokeratins KRT14 and KRT5[1,2] and transcription factors SOX2[3] and p63[4], concomitant with induction of KRT4, KRT13, and Involucrin in early differentiation (suprabasal cells)[5,6], then Filaggrin and Loricrin in late differentiation (superficial cells)[7,8]. SCD is coupled to the cell cycle in the esophagus with proliferation restricted to basal cells in the mouse[9,10]. Interestingly, proliferation in the human esophagus has been noted in the first layer of suprabasal cells, potentially due to the presence of a quiescent basal cell layer and actively cycling parabasal cell layer[11,12]. Presently, controversy exists regarding to what degree, if any, heterogeneity exists within basal esophageal keratinocytes in both human and mouse[13–20].

Age represents a well-established risk factor for development of esophageal lesions, both premalignant and malignant. Recent studies further demonstrate age-associated remodeling of esophageal epithelium via expansion of clones with mutations in cancer driver genes[21,22]. These genetic events become highly prevalent among physiologically normal human esophageal epithelium with age, despite a lack of gross alterations in tissue histology[21,22]. Although these studies provide valuable insight into the impact of tissue aging upon the mutational spectrum of esophageal epithelial cells, how aging influences the cellular landscape of epithelium at the transcriptional level has yet to be elucidated.

Here, we implement single cell RNA-Sequencing (scRNA-Seq) to provide a survey of murine esophageal epithelium in young and aged mice revealing 11 transcriptionally distinct epithelial cell populations: 6 basal, 1 suprabasal, and 4 superficial. We then perform in-depth characterization of pathways associated with both the 3 stages of lineage commitment and the individual cell populations comprising the basal, suprabasal and superficial compartments. We continue to evaluate the impact of tissue aging upon the representation and transcriptional profiles of the 11 identified murine esophageal epithelial cell populations demonstrating that mitochondrial dysfunction is a feature of aged esophageal epithelium. Assessment of the 11 identified murine esophageal cell populations in scRNA-Seq data from three-dimensional (3D) murine esophageal organoids reveals marked conservation of epithelial heterogeneity. By contrast, only 6 of the 11 cell populations identified in murine esophageal epithelium are recapitulated in human biopsy specimens. We finally utilize pseudotemporal projection in the 11 epithelial cell populations to delineate cell trajectories in murine esophageal epithelial cells as they traverse the basal-suprabasal-superficial continuum during squamous differentiation. In sum, these data provide a comprehensive analysis of the cellular and molecular heterogeneity of murine esophageal epithelial cells in the context of homeostasis and tissue aging.

## Results

**Identification and molecular characterization of cell populations in murine esophageal epithelium**. We utilized scRNA-Seq to investigate the molecular heterogeneity of murine esophageal epithelium at the level of single cell resolution. Esophageal muscle layers were peeled from dissected esophagi and resulting epithelial-enriched tissue sheets (Supplementary Fig. S1) from 5 young (≤3 months) and 5 aged (≥19 months) mice were subjected to scRNA-Seq using the 10X Genomics platform (Fig. 1a). To maximize input data from initial studies, each sample from young and aged mice were integrated for dimensionality reduction. This

minimized age-based effects as well as intra- and inter-sample variability and ensured that similar cell types across age groups were grouped in the same clusters. Additionally, integration enabled direct comparisons of the representation and differential gene expression within clusters in young and aged mice. Seurat's unsupervised dimensionality reduction and clustering workflow with Uniform Manifold Approximation and Projection (UMAP) of the 44,679 analyzed cells revealed 11 epithelial cell populations with distinct transcriptional profiles (Fig. 1b, c; Supplementary Figs. S2–S4).

To reveal sources of heterogeneity within the data, calculated UMAP dimensionality reduction was projected onto the entire epithelial dataset and loadings were determined for all genes. Among the genes that most significantly impacted cell population identification were Krt14 and Krt5 (encoding Cytokeratins 14 and 5) as well as Fabp5 (encoding Fatty acid binding protein), Krt13, Krt4 (encoding Cytokeratins 13 and 4) and Krtdap (encoding Keratinocyte differentiation-associated protein) (Fig. 1d) and we utilized these markers collectively to identify basal, suprabasal, and superficial cells within our dataset (Fig. 1e, f). RNA-FISH for Krt5 and Krtdap confirmed the expected localization of these genes in situ (Fig. 1g).

Examination of known molecular features associated with basal and differentiated esophageal keratinocytes supported our population classifications. In addition to Krt5 and Krt14, the putative basal cell markers Krt15, Sox2, and Trp63 displayed marked differential gene expression when comparing esophageal epithelial subsets identified as basal cells to those defined as suprabasal or superficial (Fig. 2a). Additionally, Mki67 expression was most abundant in population basal 2 with limited expression in basal 1 and basal 3 (Fig. 2a), consistent with restriction of proliferation to the basal cell compartment in murine esophageal epithelium. With regard to known markers of squamous differentiation, Krt4, Krt13, Krtdap, and Lor (encoding Loricrin) expression was low in basal cell populations with induction becoming apparent in superficial cells (Fig. 2a) while only a limited number of cells in the population superficial 4 exhibited expression of Flg (encoding Filaggrin) or Ivl (encoding Involucrin) (Fig. 2a).

Unbiased determination of transcripts displaying differential expression across the three stages of lineage commitment was also performed in murine esophageal epithelium with various established markers for basal, suprabasal and superficial subsets identified (Supplementary Fig. S5). Additionally, several genes that have yet to be associated with the squamous differentiation gradient in the mouse esophagus were identified (Supplementary Fig. S5). Immunohistochemical staining provided validation of COL17A1, ATP1B3 and CNFN as markers of basal, suprabasal and superficial cells, respectively, in murine esophageal epithelium (Fig. 2b). Ingenuity Pathway Analysis (IPA) of differentially expressed genes (DEGs) in the collective basal, suprabasal, and superficial cell populations further provided insight into the dynamic molecular signatures associated with lineage commitment in esophageal keratinocytes (Fig. 2c; Supplementary Fig. S6). IPA predicted activation of the unfolded protein response (UPR) in superficial cells as compared to basal cells. Indeed, UPR has previously been shown to be upregulated during and required for SCD in the esophagus[23]. As other pathways identified by IPA have yet to be implicated in esophageal differentiation to the best of our knowledge, this analysis provides a number of potential pathways to explore in relation to esophageal SCD, including eukaryotic initiation factor (EIF) 2 signaling, cholesterol biosynthesis, glutathione-mediated detoxification, glucose and fatty acid metabolism, and sumoylation. As these data are based on changes in genes expression, we sought to validate changes at the protein level in mediators of

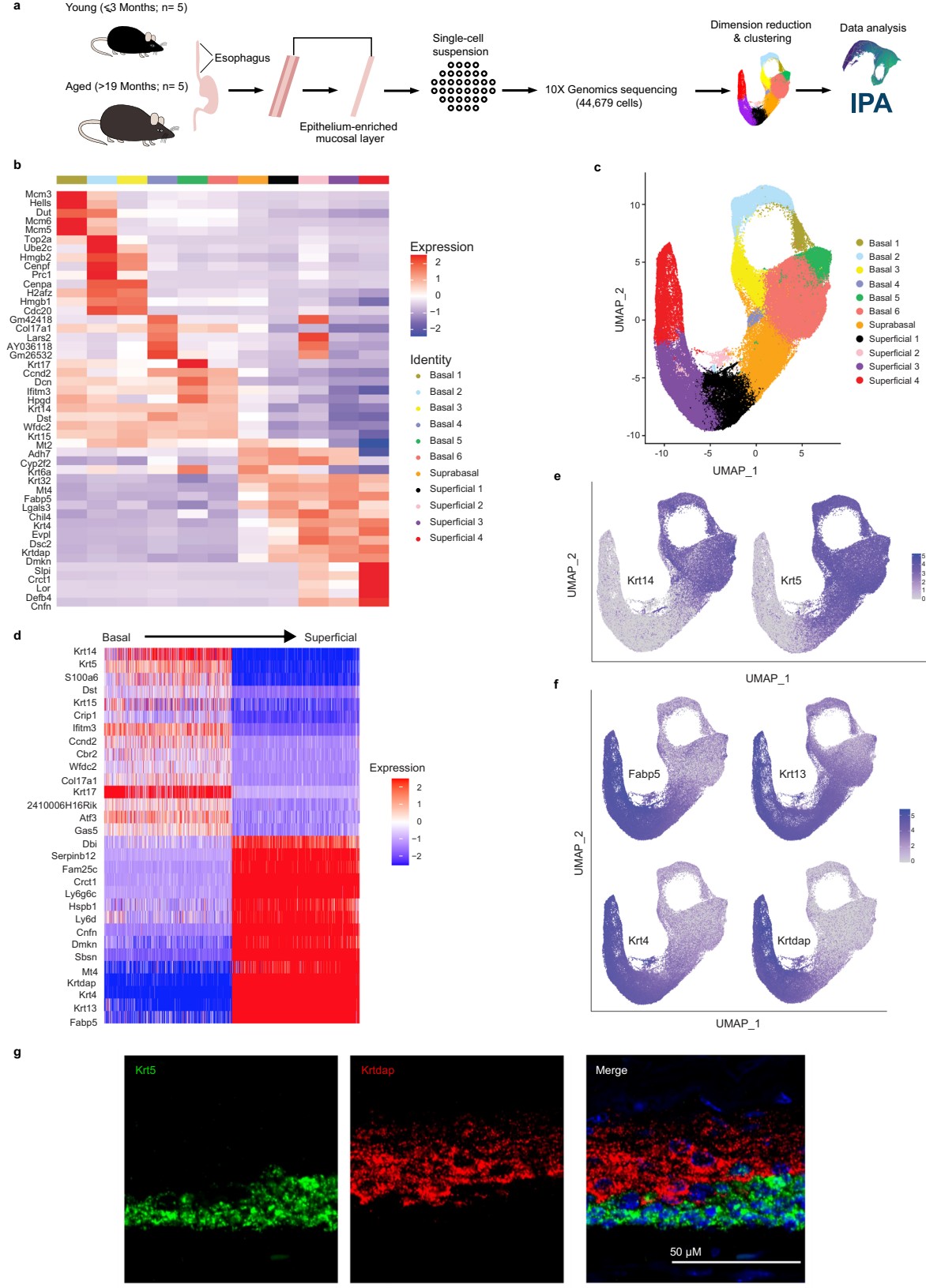

eIF2 signaling (predicted to be activated in basal cells as compared to differentiated cells) and glutathione-mediated detoxification (predicted to be activated in differentiated cells) (Fig. 2c) in murine esophageal keratinocytes undergoing SCD in

response to high calcium (0.6 mM) in vitro (Fig. 2d). Immunoblotting analysis revealed that EIF2α, EIF2Bε, RPL10, and RPS3 (associated with EIF2 signaling) are more abundant in murine esophageal keratinocytes cultured in 0.018 mM calcium whereas

**Fig. 1 Identification of cell populations in murine esophageal epithelium. a** Schematic overview of experimental design. **b** Expression z-scores for the top 5 upregulated genes in each cluster. Red indicates enrichment while blue indicates inhibition. **c** Seurat's Uniform Manifold Approximation and Projection (UMAP) was used to identify distinct cell populations within the epithelial dataset. Eleven epithelial cell populations were identified. **d** Genes identified as primary contributors to UMAP based on their loadings are listed and their expression z-scores in cells across the epithelial dataset are shown. Red indicates enrichment while blue indicates inhibition. **e, f** Log1p normalized expression of the basal markers Krt14 and Krt5 (**e**) and superficial markers Fabp5, Krt13, Krt4, and Krtdap (**f**) across the epithelial dataset is shown. Purple indicates enrichment. **g** Representative image of RNA fluorescence in situ hybridization to visualize Krt5 and Krtdap in murine esophageal epithelium in situ (*n* = 6 animals). Scale bar, 50 μm.

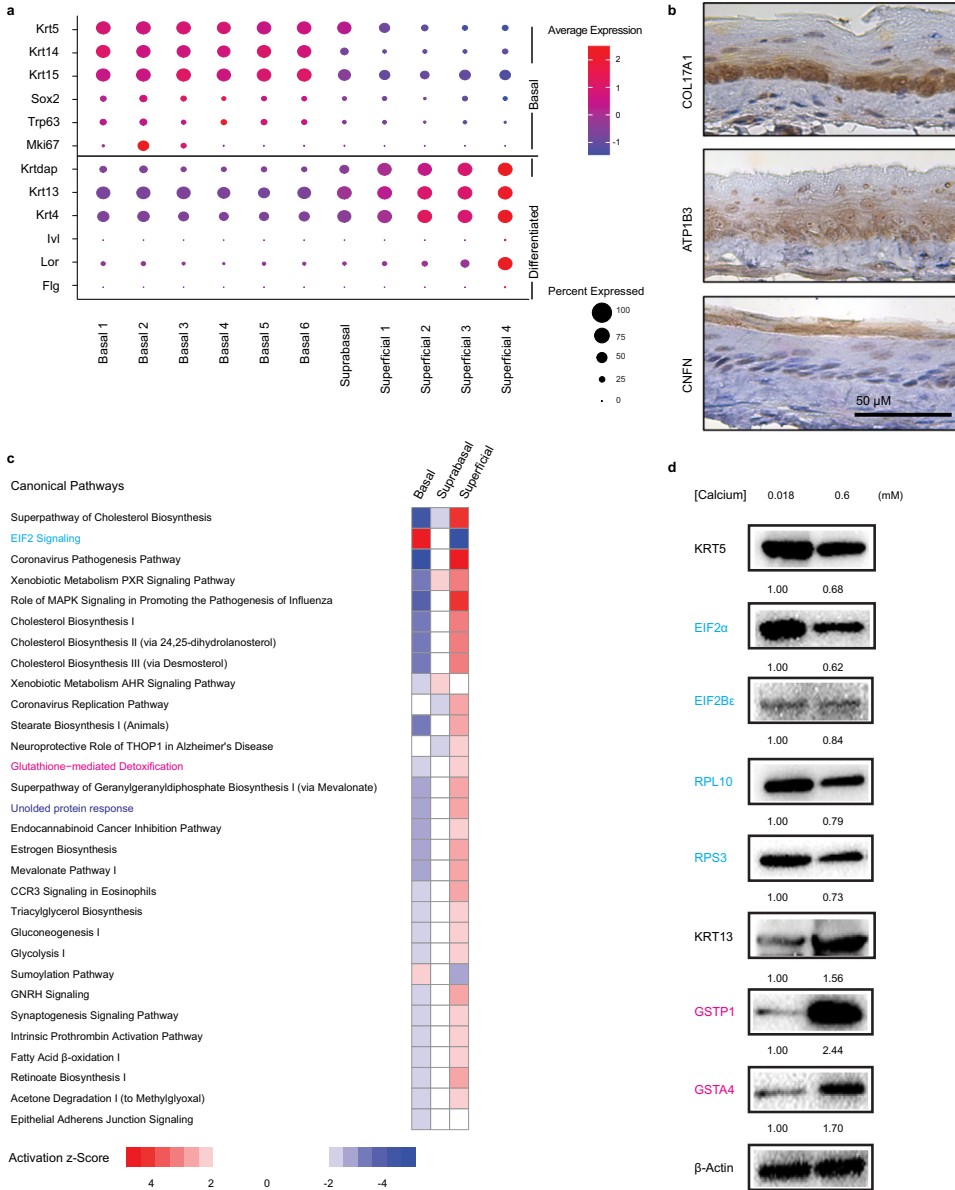

**Fig. 2 Molecular characterization of the basal/differentiated cell axis in murine esophageal epithelium. a** Cluster-average expression z-scores of putative basal and differentiated markers. Circle size reflects percentage of cells with non-zero expression level for indicated genes. Color intensity reflects average expression level across all cells within each cluster with red indicating enrichment and purple indicating inhibition. **b** Immunohistochemistry for indicated proteins in murine esophageal epithelium (*n* = 6 animals). Scale bar, 50 μm. **c** Ingenuity pathway analysis prediction of canonical pathways and their activation state in each cell types. In heat map, ranking is based upon activation z-score and color intensity depicts relative value of activation z-score. A positive value (red) indicates activation while a negative value (blue) indicates inhibition of the respective pathway. **d** Western blot analysis of representative proteins involved in eukaryotic initiation factor (EIF) 2 signaling and glutathione-mediated detoxification in primary murine esophageal keratinocytes cultured in media with indicated calcium concentrations. Densitometry determined relative level of indicated proteins normalized to β-actin. All experiments were repeated three times independently. Uncropped and unprocessed full scans are included in the source data. Source data are provided as a Source Data file.

GSTP1 and GSTA4 (associated with glutathione-mediated detoxification) are more abundant in murine esophageal keratinocytes undergoing calcium-mediated SCD (Fig. 2d).

We continued to utilize IPA to evaluate alterations in pathways, transcription factors, and kinases in the 11 individual cell populations identified in murine esophageal epithelium based upon activation z score (Fig. 3a; Supplementary Fig. S7). Among the parameters analyzed by IPA, a number displayed similar trends in prediction among cells classified as basal (e.g. pathways associated with cholesterol biosynthesis and sterol regulatory element binding transcription factor (SREBF) 1 and 2 are predicted to be inhibited in several basal populations) as well as those classified as superficial populations (e.g. pathways associated with cholesterol biosynthesis and SREBF1 and 2 are predicted to be activated in several superficial populations) (Fig. 3a, Supplementary Fig. S7a). It is important to note, however, that population-specific pathway signatures were also detected (Fig. 3b), indicating that there are unique molecular features in the individual cell populations that we have identified in the basal and superficial compartments. This finding is significant as it supports the individual molecular identity of the 11 identified populations which has the potential to be further linked to specific functional roles for each cell type.

**Defining age-associated alterations in esophageal epithelial biology**. We next aimed to investigate the impact of tissue aging upon the cellular and molecular landscape of esophageal epithelium. We did not identify age-associated alteration in the relative representation of any of the 11 murine esophageal cell populations (Fig. 4a, b). Although these data suggest that esophageal epithelial cellular heterogeneity is stable in the context of tissue aging, age-associated alterations were detected across the transcriptional profiles of the 11 murine esophageal epithelial to varying degrees (Supplementary Fig. S8). IPA revealed that the majority of pathways showing age-associated alterations in murine esophageal epithelial populations were predicted to be activated (Fig. 4c). By contrast, oxidative phosphorylation was predicted to be inhibited in basal populations 2, 5, and 6 as well as in superficial populations 1, 2, and 3 (Fig. 4c). In evaluating the 130 genes displaying age-associated differential expression in young and aged, we found that 11 were associated with mitochondrial biology (Supplementary Fig. S9). To determine whether these findings based upon gene expression analysis were relevant to mitochondrial biology, we evaluated mitochondrial DNA (mtDNA) content and complex I activity in the esophageal epithelial-enriched layer of young and aged mice (Fig. 4d, e). Increased mtDNA levels coupled with decreased complex I activity supported age-associated disruption of mitochondrial biology in esophageal epithelium of aged mice.

**Evaluation of identified murine esophageal cell populations in murine esophageal organoids and human biopsy specimens**. We next aimed to determine the relevance of the 11 cell populations identified in our murine esophageal epithelial data set in both 3D esophageal organoid culture and in human esophageal epithelium. After performing scRNA-Seq on 9,487 cells derived from murine 3D esophageal organoids and imputing cell identities established in our murine esophageal epithelial dataset (Fig. 5a, b), we found that 10 of the 11 murine epithelial cell populations were predicted to be present in 3D organoids with varying levels of transcriptional similarity among the clusters (Fig. 5c, d). Differences in the representation of several populations was also noted with decreased percentages of basal 2 and 3 as well as superficial 3 and 4 in 3D organoids where percentages of basal 5 and the suprabasal population were increased (Fig. 5e).

Notably, a cell population with transcriptional similarity to superficial population 2 was not detected in 3D organoids (Fig. 5e). We continued to explore the relationship between our murine esophageal epithelial dataset and human esophageal epithelium using a published dataset in which scRNA-Seq profiled the cellular heterogeneity within normal esophageal tissue biopsies from 9 human subjects[24]. In this dataset, human esophageal epithelium was grouped into 5 cell populations: basal, suprabasal, suprabasal dividing, intermediate, and superficial (Fig. 6a). In comparison to our findings in murine 3D organoids, there was lower degree of similarity between murine and human esophageal epithelium with only 6 of the 11 murine epithelial cell populations predicted to be present in human biopsies (Fig. 6b–d). Among these 6 populations, we detected decreased representation of basal populations 1 and 6, superficial population 3, increased representation of superficial population 4, and no change in the representation of basal populations 2 and 5 in human esophageal epithelium (Fig. 6d). Taken together, these data indicate that murine 3D organoids largely recapitulate the cellular heterogeneity present in the murine esophageal epithelium, particularly with regard to basal cells, while there are marked differences in the cellular heterogeneity present in murine and human esophageal epithelium.

**Mapping cell fate trajectories murine esophageal epithelium**. Finally, we aimed to characterize the relationships existing between individual epithelial cell populations along the proliferation/differentiation axis in murine esophageal epithelium (Supplementary Fig. S10). We employed Monocle 3 for pseudo-temporal trajectory inference mapping of the gene expression profiles (Fig. 7a) and clusters (Fig. 7b) comprising our dataset. With this technique, the overall cluster distribution found using Seurat is conserved in the Monocle 3 UMAP projection with the following notable exceptions: enhanced dispersion of basal population 5 and altered localization of basal population 4 (Fig. 7b). These differences are a result of distinct batch correction techniques employed by each algorithm with Seurat utilizing an internal integration technique[25] and Monocle 3 employing matching mutual nearest neighbors batch correction[26]. In order to construct a model of cell fate determination in murine esophageal epithelium we further evaluated cell cycle-associated genes across our data set (Fig. 7c). By integrating pseudotime projection with mapping of cell clusters and cell cycle-associated genes we developed a model wherein root cells located in the portion of basal population 6 located adjacent to the S phase-enriched cell fraction move into the cell cycle, giving rise to basal populations 1, 2, and 3 sequentially (Fig. 7d). As cells exit the cell cycle, they return to the G0/G1-enriched basal 6 pool which represents a decision point at which cells may either re-enter the cell cycle or remain in G0/G1 phase (Fig. 7d). Of the cells remaining in G0/G1, terminal differentiation appears to be the trajectory that most cells follow, moving from the basal population 6 pool to the suprabasal population and then sequentially through superficial populations 1, 3, and 4 with pseudotime failing to resolve the trajectory of superficial population 2. (Fig. 7d). Interestingly, a subset of G0/G1 appear to represent a third cell fate that is distinct from either murine esophageal basal cells that are destined to undergo cycling/self-renewal or to undergo SCD (Fig. 7d). This 'alternative cell fate' trajectory is enriched for a subset of basal population 6 and also in cells comprising basal populations 4 and 5 (Fig. 7d). We continued to examine gene modules in the pseudotemporal projection of our dataset (Fig. 7e). Of particular interest is module 18, which represents a set of genes displaying robust enrichment in basal populations 4, 5 and 6. IPA analysis of the genes in module 18

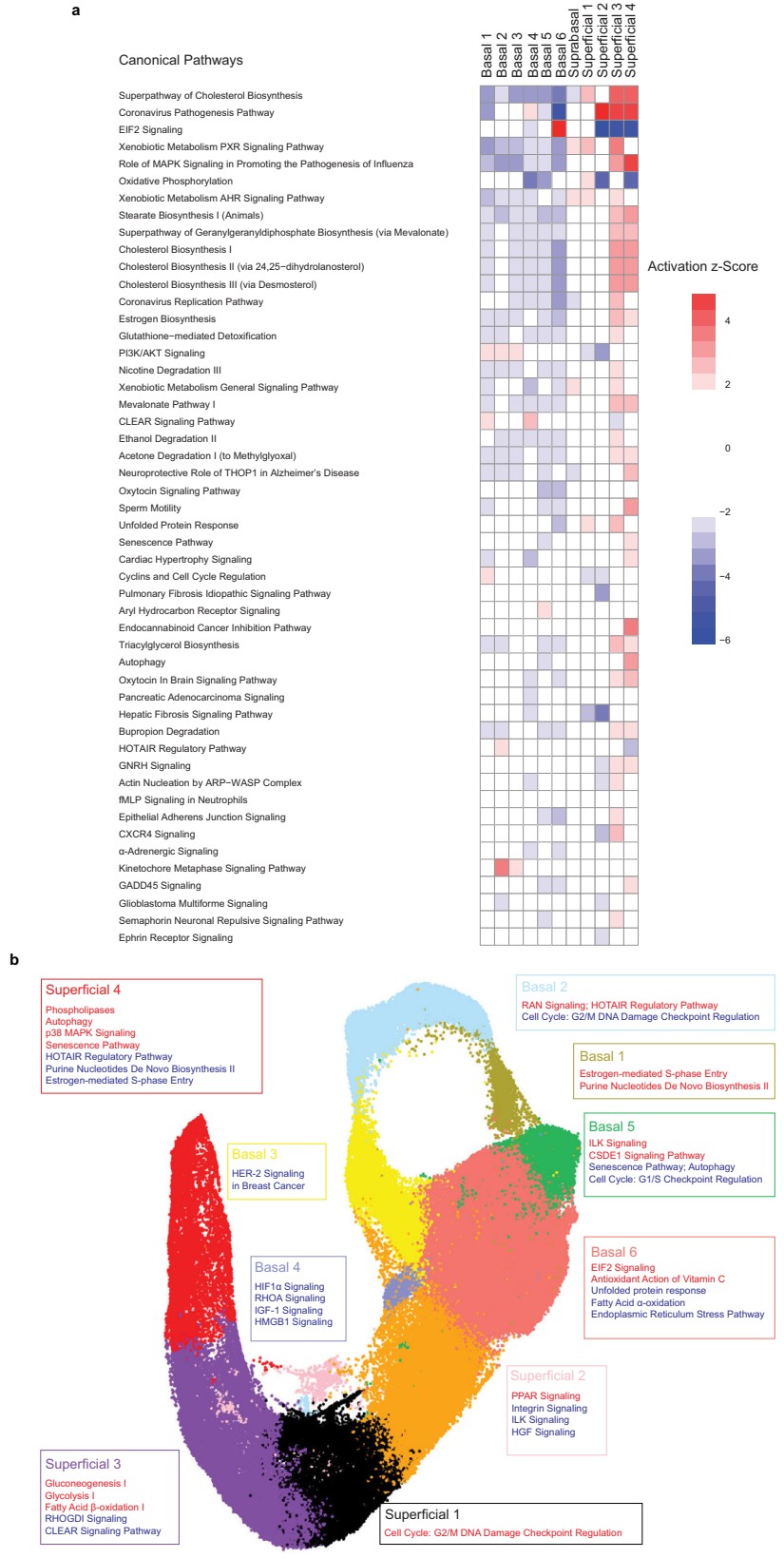

**Fig. 3 Molecular characterization of the individual cell populations identified in murine esophageal epithelium. a** Ingenuity pathway analysis (IPA) prediction of canonical pathways and their activation state in each cluster. In heat map, ranking is based upon activation z-score and color intensity depicts relative value of activation z-score. A positive value (red) indicates activation while a negative value (blue) indicates inhibition of the respective pathway. **b** Representative unique pathways identified by IPA in each population in murine esophageal epithelium along with their activation state. Source data are provided as a Source Data file. Source data are provided as a Source Data file.

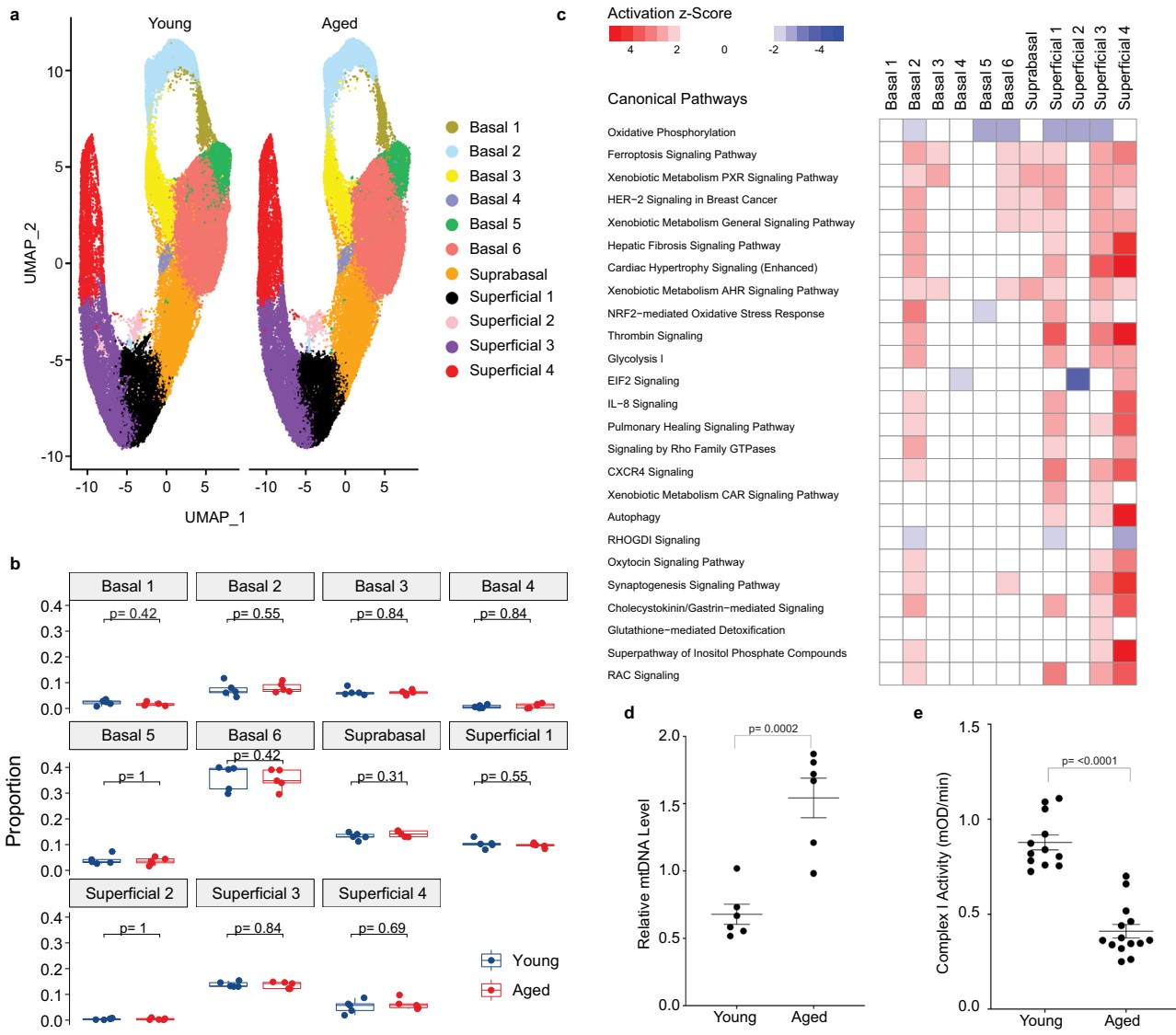

**Fig. 4 Age-associated alterations in murine esophageal epithelium. a** Uniform Manifold Approximation and Projection (UMAP) visualization of Seurat cell populations in young and aged esophageal epithelium. **b** Proportion of each cell population as a fraction of all cells in the epithelium for each age group ($n = 5$ animals per age group). Indicated $p$ values were determined using Wilcoxon signed-ranked test without adjustment for multiple comparisons. $\log_2$ of the odds ratio of fractional proportions of each cell population. Odds ratio calculated by fractions in each respective cluster in aged mice over young mice. Each individual scatter point represents proportion indicated, box indicates quartiles, whiskers indicate minima and maxima. Mean is indicated by line striking through box. **c** Ingenuity pathway analysis identified the canonical pathways significantly altered in esophageal epithelial cell populations in aged mice relative to young mice. A positive value (red) indicates activation while a negative value (blue) indicates inhibition of the respective pathway. **d** DNA level of mitochondrial D-loop was determined relative to nuclear DNA-encoded gene Ikbβ in epithelial-enriched mucosal layer from young ($n = 6$ animals) and aged ($n = 6$ animals) C57B6 mice. *$p = 0.0002$. **e** Mitochondrial complex I activity was measured in esophageal tissue lysates from young ($n = 12$ animals) and aged ($n = 14$ animals) mice. *$p < 0.0001$. In (**d**, **e**) two-tailed $t$ test was used to compare means and data are represented as mean ± SEM. Source data are provided as a Source Data file.

predicted alterations in several pathways, including Wnt/β-catenin signaling and regulation of epithelial mesenchymal transition (EMT), suggesting that these factors may contribute to alternative cell fate trajectory in murine esophageal epithelium.

## Discussion

There is a growing body of literature in which scRNA-Seq technology is leveraged for molecular characterization of squamous epithelial tissues[27,28], including the normal and diseased esophagus[24,29–35]. Our study represents a significant advance in the field of squamous epithelial biology, analyzing the transcriptome of 44,679 esophageal keratinocytes to resolve the cellular landscape of normal murine esophageal epithelium. We

further provide in-depth molecular characterization of the collective profiles of basal, suprabasal, and superficial cells as well as the 11 individual cell populations identified in murine esophageal epithelium. Such characterizations are hypothesis-generating, identifying candidate pathways, transcription factors, and kinases that may be explored in subsequent functional investigations with regard to their roles in regulating SCD in the esophagus as well as in establishing and/or maintaining the identities of the 11 epithelial cell populations. For example, the downregulation of pathways associated with metabolism of nutrients, including cholesterol, glucose, and fatty acids, are predicted to be upregulated in superficial cells as compared to basal cells. Should differences in the metabolic activity of basal and superficial cells

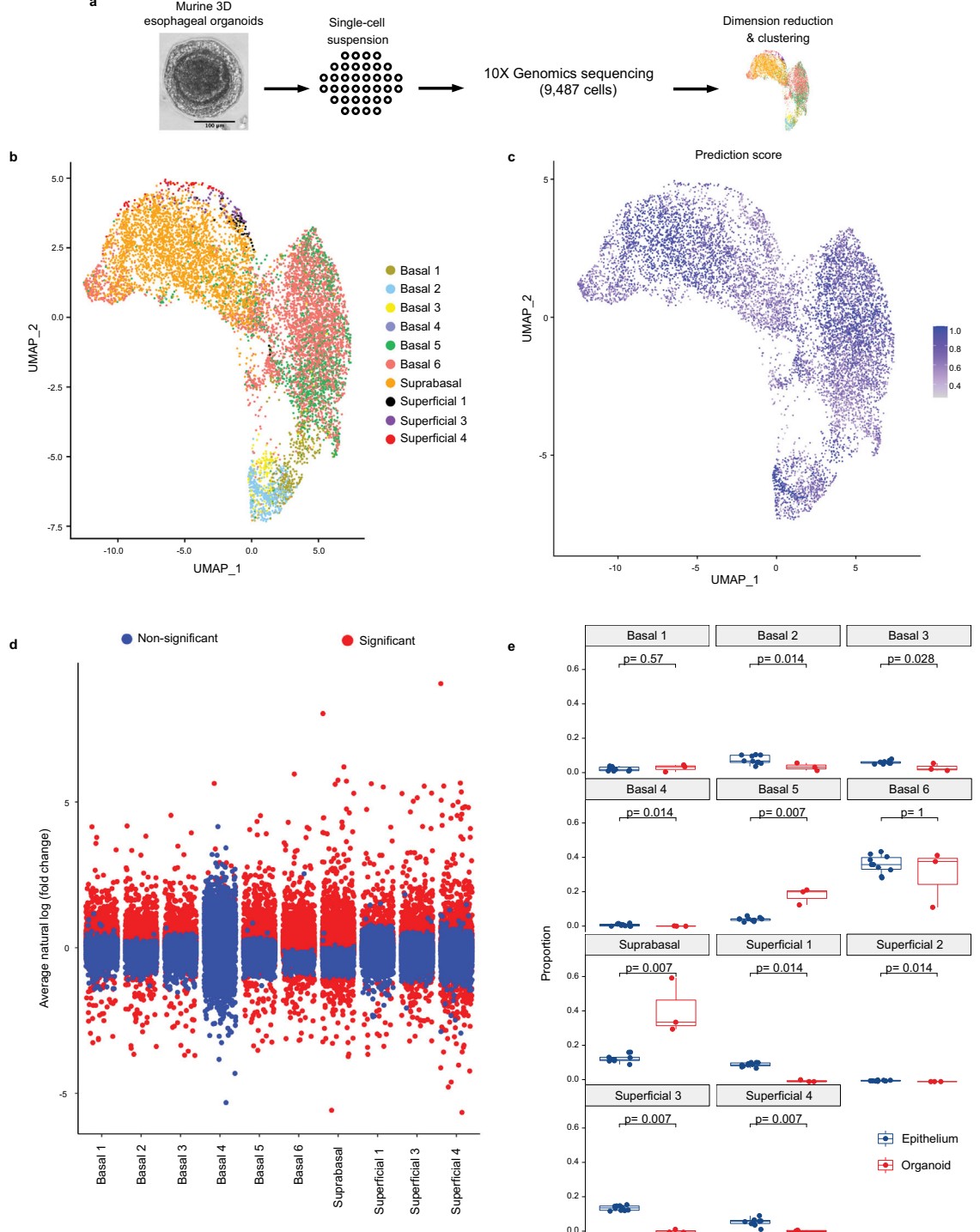

**Fig. 5 Imputation of cell populations identified in murine esophageal epithelium in 3D murine esophageal organoids. a** Schematic overview of experimental design of single cell RNA-Sequencing (scRNA-Seq) from murine esophageal organoids established from mice ($n = 3$ independent experiments) and cultured for 15 days. Scale bar, 100 μm. **b** Seurat's Uniform Manifold Approximation and Projection (UMAP) was used to visualize cell clusters within the epithelial dataset. **c** Prediction scores (based on confidence scale ranging from 0 to 1) of each cell population identified in organoids using classifications derived from murine esophageal epithelial scRNA-Seq dataset. Color intensity reflects the relative confidence that the predicted cluster identity is accurate with purple indicating cells with highest prediction confidence. **d** Average natural log fold change (FC) of individual genes in indicated cell population from murine 3D organoids as compared to the murine esophageal epithelial dataset. Genes marked as significant (indicated in red) have $p < 0.05$. **e** Comparison of proportions of each epithelial cluster in murine epithelial dataset ($n = 10$ animals) and 3D organoids ($n = 3$ independent experiments). Indicated $p$ values in (**d**, **e**) were determined using Wilcoxon signed-rank test without adjustment for multiple comparisons. Each individual scatter point represents proportion indicated, box indicates quartiles, whiskers indicate minima and maxima. Mean is indicated by line striking through box. Source data are provided as a Source Data file.

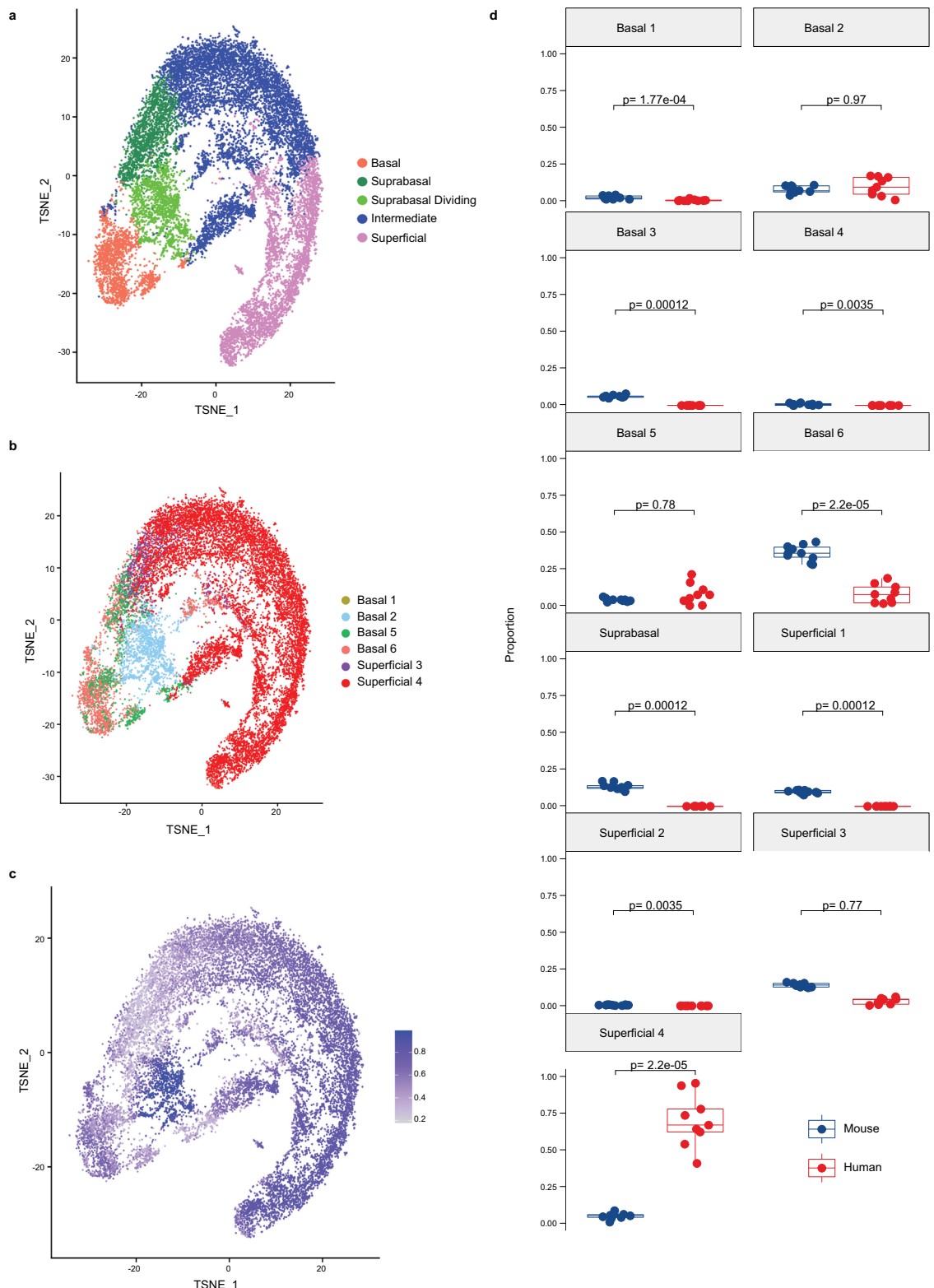

be validated, a critical consideration as our findings are based upon transcriptomic profiling, it will be of interest to determine whether they are merely a reflection of differential energy demands or if such alterations may actively contribute to SCD. While metabolic reprogramming has been linked to cell differentiation in various cell types, including mesenchymal stem cells[36,37], it has not been explored in the context of SCD in the esophagus.

In addition to shared molecular features based upon stage of lineage commitment, IPA analysis further predicts that each of the 11 cell populations identified in murine esophageal epithelium displays unique pathway alterations. Although these findings will require further validation, they support the notion that the identified cell populations may have discrete roles in the esophagus. Defining the functional significance of the 6 identified basal cell populations is of particular interest as the question of

**Fig. 6 Imputation of cell populations identified in murine esophageal epithelium in human esophageal epithelial dataset. a** Seurat's t-distributed Stochastic Neighbor Embedding (tSNE) was used to visualize the cell clusters classified by Nowicki-Osuch and Zhang, et al.[24] in the human esophageal epithelium. **b** tSNE visualization was used to visualize cell identities in human esophageal epithelial cells that are most comparable to the populations identified in the primary murine esophageal epithelial dataset. **c** Prediction scores (based on confidence scale ranging from 0 to 1) of each cell identified in human esophageal epithelium using classifications derived from murine esophageal epithelial single cell RNA-Sequencing dataset. Color intensity reflects the relative confidence that the predicted cluster identity is accurate with purple indicating cells with highest prediction confidence. **d** Comparison of proportions of each epithelial population in murine epithelial dataset ($n = 10$ animals) and human epithelial dataset ($n = 9$ human subjects). Indicated $p$ values were determined using Wilcoxon signed-rank test without adjustment for multiple comparisons. Each individual scatter point represents proportion indicated, box indicates quartiles, whiskers indicate minima and maxima. Mean is indicated by line striking through box. Source data are provided as a Source Data file.

what degree, if any, heterogeneity exists among esophageal basal cells remains controversial. Lineage tracing in mice coupled with mathematical modeling support a single-progenitor model wherein all esophageal basal cells have equal capacity to proliferate or differentiate to facilitate tissue renewal[16,20]. However, several other studies have identified markers associated with functional heterogeneity in the mouse esophagus. Slow-cycling/long-lived esophageal basal cells with self-renewal capacity have been identified by positivity for CD34, KRT15, or a combination of high integrin a6 and low CD71 expression[14,17,18]. By contrast, work by DeWard et al. indicates that a proliferative subset of basal cells defined by high expression of integrin β4 and positivity for CD73 exhibits the greatest stem cell potential in murine esophageal epithelium[15]. Herein, we integrate pseudotemporal trajectory inference mapping with expression of cycle-associated genes in the 11 cell populations identified in murine esophageal epithelium to develop a model of cell fate trajectories. We propose that G0/G1 cells in basal population 6 give rise to a pool of cells that may continually move through the cell cycle (termed 'cycling/self-renewal'), providing the capacity for tissue renewal. Such as population is consistent with the rapidly cycling progenitor population identified in the basal layer of murine oral epithelium[27]. After exiting the G2/M phase of the cell cycle, cells stemming from basal population 6 may instead elect to remain in G0/G1 with two possible trajectories: terminal differentiation or an alternative cell fate. Studies in both murine esophageal and oral epithelium support then notion that squamous epithelial cells in the basal cell layer may either remain in the cell cycle, acting as proliferative progenitor cells, or exit the cell cycle as they prepare to migrate away from the basal layer and initiate terminal differentiation[16,27]. What remains to be determined, however, is the biological significance of the alternative cell fate trajectory identified in esophageal epithelium. The end point of this trajectory is comprised of basal clusters 4, 5, and 6 and is associated with enrichment for Wnt/β-catenin signaling and regulation of EMT, both of which have been associated with stemness in esophageal epithelium[15,38–40]. Though it is tempting to speculate that this alternative trajectory may represent cells in the G0/G1 phase of the cell cycle with increased stem potential, perhaps consistent with a slow-cycling or long-lived cell population, functional studies are necessary to investigate this hypothesis. Data from the current study will serve as a valuable resource to facilitate identification of specific markers of the 6 individual basal cell populations in murine esophageal epithelium. Subsequent lineage tracing and ablation strategies may then be used to define specific roles for these cell populations in the context of esophageal homeostasis as well as disease models.

Our findings further indicate that esophageal organoids, which have been increasingly employed to study esophageal biology in an ex vivo setting[15,17,41–47], may prove to be a valuable tool in defining the functional roles of individual esophageal epithelial populations. Indeed, 10 of the 11 cell populations identified in murine esophageal epithelium, including all 6 basal populations,

are predicted to be represented when murine esophageal keratinocytes are cultured in this 3D experimental platform. Superficial population 2 is the only population that does not appear to be represented in 3D organoids. Notably, pseudotemporal trajectory inference in our murine epithelial dataset failed to incorporate superficial population 2 into the projected trajectory, revealing that cells in the suprabasal population moved sequentially through superficial populations 1, 3, and 4, the latter of which is the most terminally differentiated population. Although this finding raises questions regarding the significance of superficial cluster 2 with regard to SCD in the esophagus, it also indicates that cell populations making up the typical trajectory for cells undergoing SCD are preserved in 3D organoids.

Despite notable differences when comparing murine and human esophageal epithelium, including increased number of cell layers and the lack of luminal keratin deposition as well as the presence of papillae and a largely quiescent basal cell layer in the human esophageal epithelium[11,12], the mouse is commonly used as a model system to study esophageal biology. Here, we report marked inter-species variability with regard to cellular heterogeneity when comparing human and murine esophageal epithelium. Our data indicate that only 6 of the 11 cell types identified in mouse esophageal epithelium are predicted to be present in normal human esophageal epithelium. Notably, these 6 cell types account for all phases of the cell cycle with S phase-enriched basal population 1, G2/M phase-enriched basal 2, as well basal populations 5 and 6 and superficial populations 3 and 4, all four of which are G0/G1 phase-enriched. Consistent with the presence of a quiescent basal cell layer and proliferative suprabasal cell layer in humans, our data indicate the population defined as 'basal' in human esophageal epithelium is comprised largely of G0/G1-enriched basal populations 5 and 6 while the 'suprabasal dividing' population in humans is comprised of basal populations 1 (S-enriched) and 2 (G2/M-enriched). The human 'suprabasal dividing' and 'superficial' populations further display highest level of transcriptional similarity to their murine counterparts as indicted by prediction scoring. Thus, although our studies indicate decreased cellular heterogeneity in human esophageal epithelium relative to its murine counterpart, there is conservation of both the cell cycle and the basal/superficial cell axis. Additionally, the presence of a COL17A1 as a marker of basal cells in murine esophageal epithelium is consistent with studies in human esophageal epithelium[11,29]. Further studies are required to determine if the molecular signatures associated with basal and superficial cells in murine esophageal epithelium are conserved in human esophageal epithelium. Should this prove to be the case, it would support the potential translational utility of findings related to esophageal biology in murine models.

An additional strength of the current study is the evaluation of the impact of tissue aging upon the cellular and molecular landscape of murine esophageal epithelium. Our findings indicate that representation of the individual 11 cell populations comprising mouse esophageal epithelium is stable in the context of

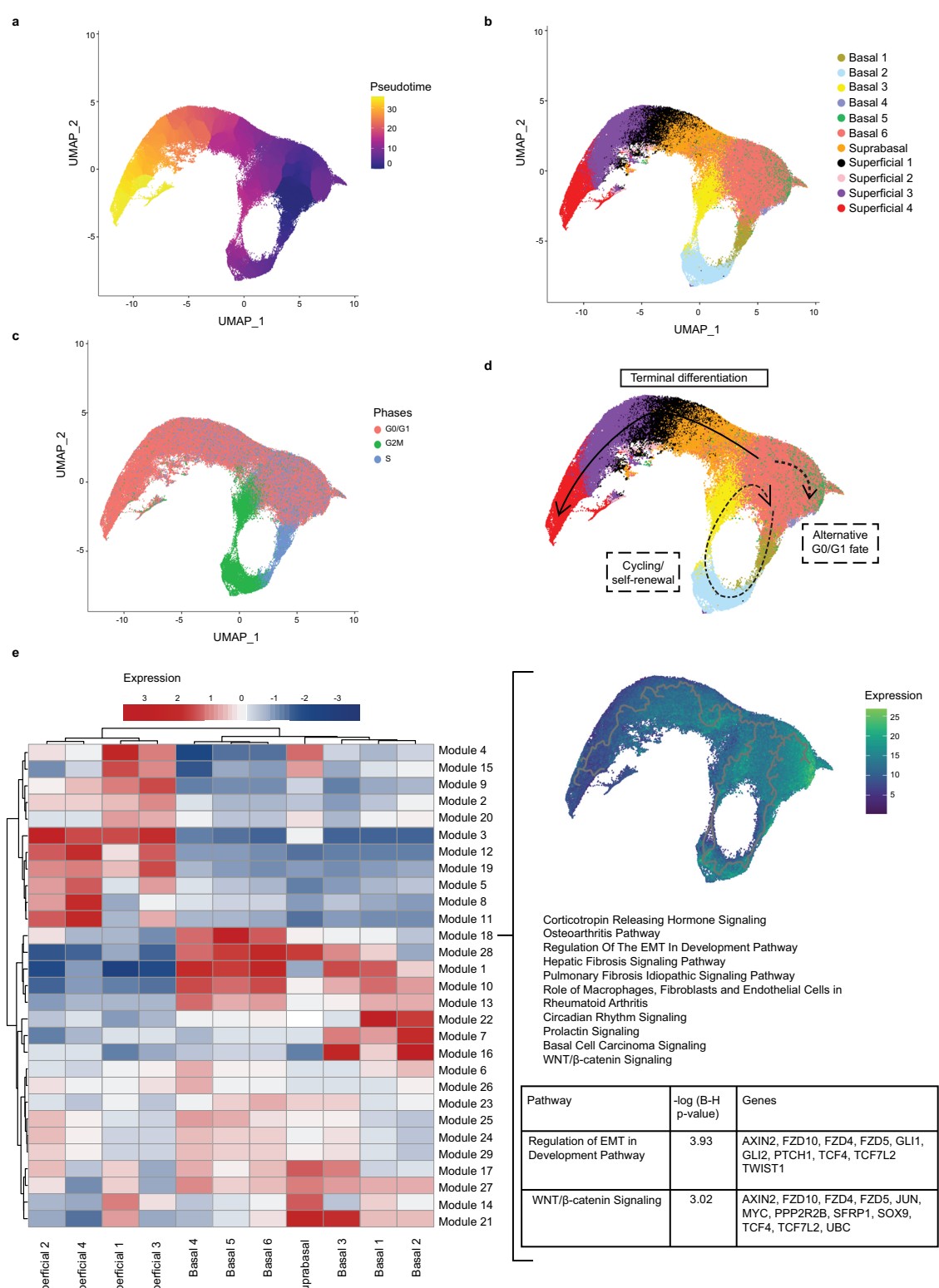

aging; however, the transcriptional profiles in these subsets are altered. In esophageal epithelium of aged mice, IPA predicted inhibition of oxidative phosphorylation and analysis of complex I activity supported impaired mitochondrial function. mtDNA level, however, was found to be increased in aged esophageal epithelium, potentially as an attempt to restore mitochondrial function. IPA further predicted enrichment of several stress-associated pathways that have been demonstrated to intersect with mitochondrial biology, including ferroptosis signaling pathway, NRF2-mediated oxidative stress, and autophagy. It remains to be determined how such alterations may play a role in age-associated esophageal diseases, including cancer.

In sum, this study provides in-depth analysis of the cellular and molecular landscape of murine esophageal epithelium under homeostasis and in the context of tissue aging. In addition to the noted future directions, it will be of interest to examine the spatial

**Fig. 7 Pseudotemporal projection of cell populations and proposed model of cell fate in in murine esophageal epithelium. a** Monocle 3's Uniform Manifold Approximation and Projection (UMAP) visualization of murine epithelial cells in the dataset. Each cell is colored by its inferred pseudotime value with dark purple representing the earliest cells and bright yellow representing the latest cells in the trajectory. **b** Clusters identified in in the integrated analysis of the epithelium labeled in the pseudotime projection. **c** Expression of genes associated with each phase of the cell cycle were labeled in pseudotime projection. **d** A proposed model of cell fate trajectories in murine esophageal epithelium. **e** Modules of co-expressed genes and their expression in the 11 cell populations identified in murine esophageal epithelium. A positive value (red) indicates activation while a negative value (blue) indicates inhibition of the respective pathway. Module 18 is highlighted because of its enrichment in basal clusters 4, 5, and 6, which comprise the proposed alternative cell fate in murine esophageal epithelium. Within Module 18, relative gene expression level is indicated from lowest (dark blue) to highest (light green). Significantly enriched pathways as identified by Ingenuity pathway analysis are listed along with genes in two pathways of interest, Regulation of epithelial mesenchymal transition (EMT) in Development Pathways and Wnt/β-catenin Signaling.

organization and interact ions of the 11 cell types identified as well as how these parameters may be influenced by aging and exposure to stimuli relevant to esophageal diseases. Such studies will build upon the cellular roadmap for murine esophageal epithelium established in the current study and further our understanding of mechanisms of homeostasis and disease in this organ.

## Methods

**Ethical considerations**. All research for the current study complies with all relevant ethical regulations. All murine studies were performed under a Temple University Institutional Animal Care & Use Committee-approved protocol (#5018).

**Murine epithelial tissue collection and processing**. Wild type C57Black6 mice (Cat# 000664) were purchased from Jackson Laboratories at age 10 weeks or 70 weeks. Mice were allowed to acclimate for at least 2 weeks prior to use for experiments. Mice were housed in individually ventilated caging racks on corncob bedding with 12:12 light-dark cycle. Mice had ad libitum access to filter-sterilized water and standard irradiated chow. Cages were changed every 2 weeks. Temperature was maintained at 68–72° F. Humidity was maintained between 30% and 70%. Whole esophagi were dissected from young (≤4 months; Range 12–13 weeks) and aged (≥19 months; Range 19–20 months) mice. For experiments using epithelium-enriched mucosal layer, muscle layer was physically removed using forceps then the esophagus was cut open longitudinally to expose epithelial surface. For single cell isolation, peeled esophageal epithelium of 5 young (12–13 weeks of age; 2 male, 3 female) and 5 aged (84–85 weeks of age; 2 male, 3 female) mice were incubated in 1 ml of 1X Dispase I (Corning 354235) in Hank's Balanced Salt Solution (HBSS) (Gibco 14025-076) containing penicillin/streptomycin (1% v/v Gibco 15140-122), gentamycin (5 µg/ml, Apex 25–533), Fungizone (500 µg/ml, Genesis 25–541) for 10 min at 37 °C with shaking at 1,000 RPM (ThermoMixer F1.5 Eppendorf). Following removal from Dispase I, esophageal epithelium was chopped into 3 pieces with sharp scissors then incubated in 1 ml of 0.25% Trypsin-EDTA for 10 min at 37 °C with shaking at 1000 RPM. Trypsin and tissue pieces were forced through a cell strainer (70 µm) into a 50 ml conical tube containing 4 ml of soybean trypsin inhibitor (STI). Cells were pelleted at 1200 RPM for 5 min and pellets were then resuspended in 500 µl of complete mouse keratinocyte–serum-free medium (Gibco Cat# 37010022). Cell number and viability were measured by Automated Cell Count (Invitrogen Countess II FL) by mixing 10 µl of cell suspension with 10 µl 0.4% trypan blue solution (1:1). For single-cell experiments, at least 300,000 cells were isolated from each mouse, serving as individual biological replicates. Dead cells were removed by Miltenyi Biotec dead cell removal kit (Cat# 130-090-101) and OctoMACS starting kit (130-042-108) according to manufacturer's instructions. Cells with 80–95% viability were used for single cell encapsulation. For downstream molecular studies, epithelium-enriched mucosal layer or whole esophagus were processed as described below.

**scRNA library preparation and sequencing**. The single cell droplets were generated with chromium single-cell controller using Chromium Next GEM Single Cell 3' Kit v3.1 (10x Genomics, cat# 1000121). 5000–7000 cells were collected to make cDNA at the single cell level. Full-length cDNA with UMI was synthesized via reverse-transcription in the droplet. After PCR amplification and purification, cDNA was fragmented to around 270 bp and the Illumina adapters with index were ligated to the fragmented cDNA. After PCR, purification, and size selection, the single cell RNA libraries were 450 bp in length and sequenced on Illumina sequencer at R1 = 28 bp, R2 = 91 bp.

**Deconvolution of scRNA-seq reads**. Deconvolution of scRNA-Seq reads followed the 10X Genomics Cell Ranger (v6.0.0) pipeline[48]. Massively parallel digital transcriptional profiling of single cells was performed using the command 'cellranger count with FASTQ files' as input from each sample. For cellranger count, R1 and R2 were trimmed to 28 and 91 bp, respectively, to remove PCR adapters. The mouse genome mm10 (GENCODE vM23/Ensembl 98) was used as the reference

for genome alignment and feature counting. From the output, the filtered matrices are used for downstream analyses.

**Data filtering, integration, dimensionality reduction, and clustering**. The matrices for each murine peeled epithelium sample were imported and transformed into Seurat (v4) objects for further processing. Cells with over 10% of their transcripts consisting of mitochondrial genes, over 3,000 unique genes, and over 10,000 total UMI were excluded to remove doublets or dead/dying cells. A total of 173,396 cell reads (replicate reads from 45,003 unique cells) passed this threshold for further analyses. Analysis of the filtered matrices follows the Seurat integration workflow described by Stuart, et. al.[25] using the SCTransform function, which normalizes counts while accounting for read depth and subsequently searches for the top 2000 most variable feature per sample with the corrected counts for integration. Reciprocal PCA was then used to find integration transcript anchors between all of the matrices. Genes used for integration were ranked by the number of matrices they appear in. From this point on, dimensionality reduction used the genes and values that were pre-processed using the integration workflow. However, raw and normalized counts were stored for downstream differential expression tests. The resulting dataset was then reduced dimensionally via PCA, resulting in 30 principal components. An elbow plot was generated to see the standard deviations of each component, which verifies that the first 30 principal components contain most of the sources of variation in the dataset. To capture all the sources of variation in the dataset, all principal components were then used as input to the UMAP dimensionality reduction procedure (arXiv:1802.03426). Because of our interest in the relationship between cell cycle phases and cell fates, we opted not to regress cell cycle genes in our dimensionality reduction steps. A Shared Nearest Neighbor (SNN) graph was then constructed with the principal components of PCA by first determining the nearest neighbors for each cell and subsequently creating the SNN guided by the neighborhood overlap between each cell and its nearest neighbors. Clusters were then determined by a modularity optimization algorithm by Waltman and van Eck[49]. The initial clustering discovered non-epithelial populations that were subsequently excised, leaving in 172,283 cell reads from 44,679 unique cells identified as epithelial and used for further analyses. The remaining cells were then re-clustered with a repeat of the integration workflow described above. To choose an optimal clustering resolution, a clustering tree was generated with the R package clustree which depicts the movement of cells across clusters as resolution is increased from 0.1 to 1 with 0.1 increments. The resolution 0.2 was chosen, as it is the earliest resolution that created several clusters that were stable as the resolution was increased, as well as having a minimal number of clusters that were composed of multiple clusters from the next lowest resolution. Described dimensionality reduction and clustering procedures were also used for the murine organoid samples. Because of the presence of dead and dying cells, cells with <1500 unique genes were excluded from the dataset before dimensionality reduction and clustering. As a result, 9487 unique cells were used for dimensionality reduction and clustering for further analyses.

**Cell cluster analyses**. For each cluster, DEGs were calculated by comparing the expression of genes within the cells of the cluster over the expression of the genes in all other clusters. The statistical workflow to determine differential expression was Seurat's implementation of the Wilcoxon's signed-rank test. The significance cutoff for DEGs is a Bonferroni-adjusted $p$ value of 0.05, and the fold change cutoff is below $-0.25$ or above 0.25 natural log fold change. To characterize the differential regulation of pathways in each cluste, DEGs that pass the cutoff from each cluster were exported into "IPA [http://www.ingenuity.com]" (Qiagen) for IPA core and comparison analysis. The Seurat function CellCycleScoring was used to predict the cell cycle of each cell. The function takes as input a list of S phase upregulated genes and G2/M upregulated genes, and outputs the score for each phase. The S and G2/M phase genes were provided within the Seurat package as objects "cc.genes.updated.2019$s.genes" and "cc.genes.updated.2019$g2m.genes", respectively. The cell cycle phase is determined by the dominating score. Cells with weak scores for both phases are classified as G0/G1 phase cells. To compare each clusters' proportional size between the different age groups, each sample's cluster proportions were calculated, and a Wilcoxon signed-rank test was performed to compare the mean cluster proportions between the two age groups.

**Pseudotime**. For pseudotemporal inference, cells analyzed with Seurat were exported to Monocle 3. Pre-processing in Monocle 3 follows the methodology outlined by Cao et al.[26], which includes a batch correction step treating each mouse as a different batch to find a commonly shared reduced dimension. To verify that the cluster stratification of our primary murine epithelial dataset was reproducible, we used Monocle's default parameters without any changes. Monocle 3's procedure results in a UMAP structure, from which a trajectory graph was inferred. To model how epithelial cells cycle and assume different cell fates, the cells of basal cluster 6 prior to entrance into the S phase of the cell cycle were chosen as the root population. Finally, to further elucidate the biological processes that govern epithelial proliferation and differentiation, Monocle 3 was also used to find modules of co-expressed genes.

**Bulk RNA-seq comparison of epithelial and stromal tissue**. To identify DEGs between epithelial and stromal tissue, FASTQ files from both epithelial and stromal bulk RNA-Seq experiments were first aligned to the GRCm38.p6 mouse genome from GENCODE using the library Rsubread on R[50]. The resulting BAM files were then summarized at the gene level and counted using Rsubread's featureCounts functionality, producing a counts matrix for all the epithelial and stromal samples and their gene counts. The count matrix was then used to compare epithelial and stromal gene counts using the library DESeq2 in R[51] with an alpha of 0.05.

**In situ studies**. RNA fluorescence in situ hybridization (FISH), immunohistochemistry (IHC), and immunofluorescence (IF) were performed in formalin-fixed paraffin-embedded murine esophageal specimens. IHC was performed for COL17A1 (Invitrogen, MA5-24848l, Clone 2C3 1:100), ATP1B3 (Abcam, ab137055; Clone EPR8981, 1:100) and CNFN (Novus Biologicals, NBP2-14668; 1:100). Slides were counterstained with Hematoxylin and imaged on a Leica DM30 microscope at 400X magnification. IF was performed for KRT14 (NeoMarkers, MS-115-PABX; Clone LL002; 1:200). Slides were counterstained with DAPI and imaged at 200X magnification. IHC and IF were performed using standard protocols[39,43]. RNA FISH was performed using RNAscope technology (Advanced Cell Diagnostics) following the manufacturer's protocol and RNAscope probes for murine *Krt5*, *Krtdap*, positive control, and negative control. Slides were counterstained with DAPI and imaged on a Leica SP8 confocal microscope at 400X magnification.

**Immunoblotting**. $2 \times 10^5$ primary murine epithelial cells were seeded in 6-well plates. After 72 h, cells were treated with 0.018 or 0.6 mM $CaCl_2$ for an additional 72 h. Cells were lysed in cell lysis buffer (Cat# 9830 S, Cell Signaling Technology) containing protease/phosphatase inhibitor cocktail (Cat# 5872 S, Cell Signaling Technology). Protein concentration was determined by Qubit™ protein assay kit (Cat# Q33211, Invitrogen). Protein samples were solubilized in NuPAGE™ LDS Sample Buffer (Cat# NP0007, Invitrogen) and denatured with NuPAGE™ sample reducing agent (Cat# NP0009, Invitrogen) containing 50 mM dithiothreitol. 30 μg of denatured protein was fractionated on NuPAGE™ Bis-Tris 4–12% gel (Cat# NP0335BOX, Invitrogen). Following electrotransfer, Immobilon-P PVDF membranes (Cat# IPVH00010, Millipore Sigma) were blocked in blocking buffer containing 5% nonfat milk (Cat# LP0031B, ThermoFisher Scientific) in PBST (PBS and 0.1% Tween 20) for 1 h at room temperature. Membranes were then incubated overnight with primary antibodies diluted in blocking buffer and then with the appropriate HRP-conjugated secondary antibody for 1 h at room temperature. β-actin served as a loading control. A list of antibodies with dilutions used is provided in Supplementary Table S1.

**3D organoid assays**. Murine esophageal 3D organoid formation assays were performed on freshly isolated primary murine epithelial cells (PMECs)[43]. Briefly, a single cell suspension of PMECs in keratinocyte serum free medium was mixed with 90% Matrigel. For each well of a 24-well plate, 500–1000 cells in 50 μl Matrigel were seeded to initiate 3D organoid formation. After solidification, 500 μl of advanced DMEM/F12 supplemented with 1X Glutamax, 1X HEPES, 1X penicillin-streptomycin, 1X N2 Supplement, 1X B27 Supplement, 0.1 mM N-acetyl-L-cysteine, 50 ng/ml human recombinant EGF, and 2.0% Noggin/R-Spondin-conditioned media was added and replenished every 3–5 days. At the time of plating, 10 μM Y27632 was added to the culture medium. Organoids were grown for 15 days before recovering from Matrigel with Dispase I. Then, organoids were dissociated in 1 ml of 0.25% Trypsin-EDTA for 1 h at 37 °C with shaking at 1000 RPM. Trypsin and cells were forced through a cell strainer (70 μm) with 4 ml of 250 μg/ml STI in 1X PBS. Cells were pelleted at 1000 RPM for 5 min then resuspended in 500 μl of complete mouse KSFM. Cell number and viability were measured by Automated Cell Count.

**Imputation of cell populations in scRNA-seq data from human biopsy specimens and 3D murine**

*Esophageal organoids*. Human epithelial scRNA-Seq data was obtained from Nowicki-Osuch and Zhang et al.[24]. A reference dataset for the normal human esophagus was made publicly available on the "Esophagus Cancer Atlas [https://www.esophaguscancercellatlas.org]", from which the R object NE.rds was

downloaded. For each cell in either the murine esophageal organoid or human epithelial dataset, an imputation was done to infer which of the cell populations in our integrated murine peeled epithelium data is analogous. This is done with Seurat's label transfer pipeline (FindTransferAnchors and TransferData), which finds anchors between the reference integrated murine epithelium dataset and each query dataset and subsequently transfers the cell population labels onto the cells in the query datasets. Population labels are chosen based on which of the reference population has the max prediction score (a scale ranging from 0 to 1 signifying confidence for each label) for each of the query cells.

*Mitochondrial assays*. The activity of mitochondria Complex I was measured in peeled murine epithelium-enriched mucosal layer using Complex I Enzyme Activity Assay kit (Abcam, ab109721) according to the manufacturer's instructions. Briefly, epithelium-enriched mucosal layer from young and aged mice was suspended in 500 μl chilled PBS and completely homogenized using a Dounce homogenizer with 20–40 passes. Protein lysis buffer was added to the tissue for protein extraction followed by 30 min ice incubation to allow solubilization. Samples were centrifuged at 16,000 x *g* for 20 min at 4 °C. Supernatant was collected as tissue lysate and diluted to a desired concentration after protein estimation. Tissue lysate was added to 96-well microplates precoated with capture antibodies specific for Complex I. Once target was immobilized, Complex I activity was determined following the oxidation of NADH to NAD and the simultaneous reduction of a dye. Absorbance was measured at OD = 450 nm using a spectrophotometer. mtDNA level was measured by qPCR of DNA from epithelium-enriched mucosal layer. DNA isolation was performed using DNeasy Blood and Tissue Kit (Qiagen Cat# 69506) according to the manufacturer's instructions. qPCR was performed using PowerUp SYBR green master mix (Thermo Fisher) with the following primers: Ikbβ For: GCTGGTGTCTGGGGTACAGT Rev: ATCCTTGGGGAGGCATCTAC, and mtDNA D-Loop Fwd:AC-TATCCCCTTCCCCATTTG Rev: TGTTGGTCATGGGCTGATTA. The relative fold change between samples of mtDNA D-loop was calculated with normalization to the nuclear encoded Ikbβ.

*Statistics*. Descriptive statistics are presented as mean ± standard error of the mean or median (minimum–maximum) for continuous variables and frequency counts (percentages) for categorical variables. Two-sample *t* test or Wilcoxon rank-sum test and one-way analysis of variance or Kruskal–Wallis test comparing two and three groups, respectively for continuous variables and chi-square test or Fisher's exact test for categorical variables were used.

**Reporting summary**. Further information on research design is available in the Nature Research Reporting Summary linked to this article.

## Data availability
The authors declare that all data supporting the findings of this study are available within the article and its supplementary information files or from the corresponding author upon reasonable request. Source data are provided with this paper for the following figures: 2c, d; 3a; 4b–e; 5e; 6d; 7e; S6; S7. Gene expression and pathway information for each cluster generated in this study are provided in the Supplementary Information/Source Data file. The processed cell and gene matrices are available at GEO accession "GSE193376" as Supplementary Files. The repository Whelan-scRNA-Esophagus-Dec21 is archived on Zenodo under the https://doi.org/10.5281/zenodo.6286725 on February 25, 2022.

## Code availability
Custom scripts are available at https://github.com/alkarami/Whelan-scRNA-Esophagus-Dec21.

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

## Acknowledgements

We thank Jean-Pierre Issa, MD, and Erica Golemis, PhD, as well as members of the Fels Cancer Institute for Personalized Medicine, Fox Chase Cancer Center Histopathology Facility, and the Children's Hospital of Philadelphia Gastrointestinal Epithelium Modeling Program for technical and conceptual support. The following grants supported this work: R01DK121159 (KAW), P01CA098101 (RCA), R01DK124369 (KEH), R01DK124266 (ABM), P01DK117824 (MPT), R01DK116988 (MPT), T32GM142606 (ADF; PIs: Xavier Graña, Jonathan Soboloff, Temple University) P30CA006927 (YT, PI: Jonathan Chernoff, Fox Chase Cancer Center), Willian J. Avery Fellowship (MFK), and Charles H. Revson Senior Fellowships in Biomedical Science (RCA).

## Author contributions

Data collection and analysis: MFK, ALK, AK, RCA, AK, RS, AM, MGM, JC, ADF, MHC, KNC, YT, KK, JM, HH, JJ, KAW. Study design and interpretation: MFK, ALK, AK, KNC, JM, JJ, TK, KEH, ABM, MPT, KAW. Manuscript writing and editing: MFK, ALK, RS, MGM, ADF, TK, KEH, ABM, MPT, KAW.

## Competing interests

The authors declare no competing interests.
