## [Peer Review File · Nature Communications]

REVIEWER COMMENTS

Reviewer #1 (Remarks to the Author):

In the paper, "Single Cell transcriptomic analysis reveals cellular diversity of murine esophageal epithelium and age-associated mitochondrial dysfunction" Kabir et al. describes cellular niches in the esophageal epithelium. The paper represents interesting insights into the differentiation course of basal cells in the squamous lineage. While the discoveries are highly novel the paper leans heavily on the single cell RNA seq experiment. The paper would benefit tremendously from experimental validation of some of the results and a more unbiased approach to what is found. Further, it would be beneficial to be consistent in the type of investigations done throughout the paper. Currently, there is the feeling that some cherry picking has been done regarding the type of analysis to show the results that are most in line with the authors hypotheses. I recommend the following major revisions:

-The results from the RNA seq demonstrate that DNA metabolic processes and cell cycle regulation are heavily enriched in the clusters (Figure S5). These data are not mentioned at all in the paper that focuses on more or less random changes in single genes. It would be tremendously beneficial to focus more on the broader changes than on the individual genes. Further, these alterations should be validated in other systems for example in vitro.

-Analysis of clusters should be done in young cells only.

-All data should be made available as sortable excell sheets where it is possible to investigate the different proposed clusters.

-S5 shows if pathways are upregulated or downregulated but S6 just shows if the pathways are significantly changed. S6 should show if the pathway is upregulated or downregulated.

-S6 shows the differences in each cluster while S5 does not. S5 should show the changes in each cluster individually.

-In several cases the most interesting results are not discussed. For example, "Gene ontology (GO) analysis of differentially expressed genes (DEG) in basal, suprabasal and superficial cell clusters further provided insight into the dynamic molecular signatures associated with lineage commitment in esophageal keratinocytes (Supplementary Figure S5A-D" and "Additionally, IPA provided insight into the

pathways that are altered in the primary and minor branches (Figure 2E)". It would be of much greater interest to highlight these broader findings rather than the lengthy description of single gene changes.

-The organoid culture results are very surprising. It would be highly beneficial to be able to determine experimentally if cells from young mice have a propensity to stay stem like while old cells have a propensity to undergo differentiation. It would be further interesting to quantify if there are more or less of the different clusters of cells with age in vivo in the squamous epithelium.

-For Figure 4 it is unclear why String analysis is chosen for looking at broader gene expression changes. It would be beneficial to stick with one analysis type (GO for example).

-Altered mitochondria function is somewhat surprising given that esophageal changes are not normally seen in patients suffering from primary mitochondrial dysfunction. This should be discussed.

-Many accounts of references to previous figures in the paper. This is quite confusing.

Reviewer #2 (Remarks to the Author):

The esophageal epithelium of the mouse has been subject to intense study over the last decade. Here the authors apply single cell RNA sequencing to this tissue, seeking to establish the transcriptional states in the basal layer that contains proliferating cells and the upper layers that are undergoing differentiation. This is interesting as it builds on a wealth of transgenic modelling and the approach has proven informative in the similar stratified squamous epithelium of the oral cavity (PMID 30472156). The authors argue that there are multiple populations in the basal layer, new differentiation markers can be identified and that mitochondrial dysfunction is a feature of aging, a finding that is then followed up by functional studies.

The data comes from 3 young and 3 aged mice, each of which was analyzed in technical duplicates. This gives the opportunity to look at the technical variation within each mouse, but it appears from Fig. S1G that the two runs were just summed. It would be helpful to present the intra-mouse sample variation. Initially, all the mice are then pooled for analysis. It is notable, however, that one mouse (aged 1) contributes little data while aged 2 has a far higher proportion of mitochondrial transcripts than the other animals. It is essential to know the extent to which the mitochondrial aging phenotype described here is altered by the presence of this inter animal variation, especially given the limited contribution of

mouse 1 to the data set. Given the variability of the data, adding 2-3 more aging mice may resolve this issue. Given these issues the validity of grouping all animals together needs to be carefully justified.

Fig 2 A and B confirm that the basic structure of the tissue with basal, early and late differentiation is captured in the dataset. Immunostaining is used to confirm protein expression of selected markers on a basal to differentiated axis. Lineage reconstruction is performed in two principle components with separate analysis and clustering. Why were 2 PC selected? How robust are the clustering results to variation of the analysis parameters. Given that this is the bulk of the paper it is critical to investigate this, as the clustering may change.

The authors focus on 5 putative basal cell populations and their characteristics, exploring differences in GO terms. This is fraught with pitfalls as it depends on GO term assignments which is not necessarily reliable (https://doi.org/10.1007/978-1-4939-3743-1_14 and <https://www.nature.com/articles/s41598-018-23395-2>). For example, in Fig 2E are Col17a, Krt14, Krt15 and Krt5 really linked to cell cycle regulation and the activity of Plk or Krt4 and Krt14 to glutathione redox and the unfolded protein response? If so how? This analysis does not seem a robust way to infer biologically meaningful differences based on the genes shown and more critical assessment of this part of the study is required assuming it survives the tests of robustness above.

In Fig 3 GO terms are used to argue for the existence of a quiescent basal cell population. This is odd, as transgenic proliferation assays reveal no evidence of such a population in mouse esophagus (PMC3527005, PMC7080751). Given this data, how reliable is the assignment of quiescence based on GO analysis? It is surely essential to constrain analysis of high dimensional data using experimental data where this is available. This section should be re-evaluated in the light of the literature on the esophageal epithelium.

Figure 4 compares young and aged animals. If this analysis is to be performed more mice should be analysed as discussed above. The attribution of mitochondrial differences may reflect the marked variation in % mitochondrial transcripts across animals. The organoid data may be interesting, but needs more characterisation. The statement that 'altered basal cell dynamics...(are) a feature of esophageal aging' would not seem to be reliable given this rests on second passage organoid cultures.

The discussion is overlong and does not seek to compare the analysis here with similar epithelia in the mouse (oral and interfollicular epidermis) where functional and scRNAseq data are available.

Overall while a comprehensive scRNA seq analysis of mouse esophagus with aging would be a useful resource, this cannot be built on such a small number of animals with variable scRNAseq quality.

Reviewer #3 (Remarks to the Author):

In this paper, Kabir et al perform ssRNA sequencing of the homeostatic murine esophagus of young and aged mice. They have identified several epithelial subpopulations with distinct molecular signatures, revealed an epithelial differentiation program, and observed disbalance in epithelial differentiation particularly of the basal cells as well as impaired mitochondrial activity in aged mice. The paper provides a detailed perspective on the epithelial heterogeneity in the murine homeostatic esophagus and advances ssRNA sequencing as an important tool to further explore esophageal biology.

Comments:

1. The authors utilize esophageal organoids from young and aged mice to demonstrate disbalance in the basal cell subpopulations observed in the aged mice. However, the relevance of this system to serve as a functional readout for cellular heterogeneity in the esophageal epithelium is unclear. Which cells form the organoids? It is reasonable to assume that initially, the epithelial cells need to proliferate to form the organoid. Does this suggest that quiescent basal cluster 5 is lost? What happens during differentiation? Is heterogeneity of the epithelial cells, in particular basal cells, observed in the fully differentiated organoids? As the authors show, the proliferation rate of the organoids from the aged mice is increased with passages, which is inconsistent with the increased amount of quiescent cells from the basal layer 5 in the aged mice. To answer these questions scRNAseq of the organoids should be performed and compared to the signatures of the epithelial populations in the esophagus.

2. It is important to address fundamental differences in the esophageal epithelial biology between humans and mice in the introduction and discussion of the paper. This is especially relevant to the properties of the basal layer cells in contact with lamina propria that remain mostly quiescent in humans. The relevance of the results of the study to human vs mouse esophageal epithelial biology should be highlighted.

3. In several places in the paper, the description of the results is vague. For example, p7 line 6 "...provided insight into the dynamic molecular signatures associated with lineage commitment.." This statement refers the reader to the figure that is overloaded with data and hard to follow. Please state clearly the main findings of the analysis and label the figure accordingly. Another example is p7 line 11 related to the individual molecular identity of 8 clusters. Please detail these findings and clarify Figure 2C accordingly. Another example, page 8 line 6 "...provided insight.." What is the finding here?

4. Related to above point, it remains unclear which molecular markers separate basal subpopulations. Figure 2C does not seem to be clustered in any way and is hard to follow. Please simplify or color code relevant pathways.

5. Similarly, it is unclear how many genes were differentially expressed between aged and young mice. Was it 64 genes as stated on p11 line 15?

6. The authors use published ssRNA seq data for the human esophagus to assess the expression of the basal, suprabasal, and superficial markers identified in the paper. They conclude that these markers "display appropriate localization" in the human epithelium. I disagree with the conclusion. Based on the annotation of the data, expression of COL17A1 is not primarily basal and CNFN seems to be expressed throughout the esophageal epithelium. I am not sure how this data set confirms the specificity of the expression at least for these markers. Related to it, Col17a1 has previously been shown as a basal

marker for skin and esophageal epithelial cells (PMID: 31451683; PMID: 32187560). The differences between mouse and human also need to be considered.

7. Please clarify the significance of minor vs major branches identified in the pseudotime analysis.

8. Please label epithelial clusters in figure 3E.

9. Please explain the impact of peeling the skin layer; how specific is this to the epithelium and how sensitive to the whole epithelium? Histological imaging before and after peeling, as well as assessment of the remaining non-peeled tissue for bulk RNAseq would assess this.

10. In figure 1B, Why are the authors only averaging expression only for each cluster? Single-cell-wise expression heatmap would be informative.

11. In supplementary Figure 2, Mouse 1/2/3 seems to be biological replicates. Then what is "Replicate 1 and 2"? Figure legends says "2 reads were done for each mouse". Are they sequencing replicate or separate library preparation?

12. In figure 1, reproducibility across biological replicates are not addressed except for the Fig S1F. More quantitative assessment is needed. for example cell % in each cluster per sample. This is critical information because there can be a cell cluster originated from biological variability. Color-scheme is confusing (especially basal 2 vs 5). Direct annotation on top of the UMAP is strongly recommended.

** See Nature Research's author and referees' website at www.nature.com/authors for information about policies, services and author benefits.

RE: NCOMMS-21-04499A; 'Single cell transcriptomic analysis reveals cellular diversity of murine esophageal epithelium and age-associated mitochondrial dysfunction' by Kabir MF and Karami A, et al.

REVIEWER COMMENTS

Reviewer #1 (Remarks to the Author):

1. In the paper, "Single cell transcriptomic analysis reveals cellular diversity of murine esophageal epithelium and age-associated mitochondrial dysfunction" Kabir et al. describes cellular niches in the esophageal epithelium. The paper represents interesting insights into the differentiation course of basal cells in the squamous lineage.

We thank the Reviewer for these kind words.

2. While the discoveries are highly novel the paper leans heavily on the single cell RNA seq experiment. The paper would benefit tremendously from experimental validation of some of the results and a more unbiased approach to what is found.

We now perform experimental validation by evaluating pathways that are predicted to be differentially modulated in the basal and superficial compartments using an in vitro system in which primary murine esophageal epithelial cells are stimulated to undergo squamous cell differentiation in response to exposure to high calcium media. In revised Figure 2C, we report that EIF2 signaling is predicted to be enriched in basal cells as compared to differentiated cells while the opposite is true with regard to glutathione-mediated detoxification. We validate these findings using western blotting in the described in vitro system in Figure 2D, demonstrating that EIF2 α , EIF2B ϵ , RPL10, and RPS3 (all associated with EIF2 signaling) are more abundantly expressed in basal cells as compared to cells undergoing squamous cell differentiation whereas GSTP1 and GSTA4 (associated with glutathione-mediated detoxification) are more abundantly expressed in cells undergoing squamous cell differentiation as compared to basal cells.

3. Further, it would be beneficial to be consistent in the type of investigations done throughout the paper. Currently, there is the feeling that some cherry picking has been done regarding the type of analysis to show the results that are most in line with the authors hypotheses.

In the initial submission, we used a combination of Ingenuity Pathway Analysis (IPA), PANTHER and STRING as each of these tools has their advantages and limitations and we felt that they would provide a richer analysis of our dataset. For example, in original Figures 2C, 2E, 3B, 3D, IPA predicted alterations in many pathways in cell cycle, cell differentiation, metabolism and translation. We felt that showing GO analysis, which categorizes genes based on their association with biological processes, molecular functions etc., for these datasets would complement the IPA-analysis by providing information on the altered pathways (e.g. in terms of cell cycle, GO indicates changes in mitosis specifically.) Likewise, we utilized PANTHER, a GO-based bioinformatics tool which can predict over/under-representation of biological processes according to gene expression fold changes, to define processes that are positively or negatively enriched in our dataset (Original Supplementary Figures S5 and S6). The other tool we used is STRING, a network generation tool that creates a network based on interaction of the genes/proteins and then categorizes the GO: biological processes and pathways associated with the network. As only 64 genes significantly expressed in aged mice, we opted not to use IPA or PANTHER for which a minimum of 100 genes is recommended. However, we felt it was important to understand the differences in biology that may occur as a result of the 64 differentially expressed genes in aged mice. To investigate this, we analyzed 64 genes with STRING (Original Figure 4E). Similarly, we used STRING for molecular characterization of minor branches (Original Supplementary Figure S8) due to a smaller number of genes. We now, however, appreciate that this may have contributed to a lack of clarity in the initial submission and exclusively utilize IPA for pathway analysis in the revised manuscript. We now realize that using multiple tools for pathway analysis contributed to a lack of clarity and even concerns about cherry picking in the original manuscript. We apologize for this and now exclusively utilize IPA for pathway analysis in the revised manuscript.

I recommend the following major revisions:

4. The results from the RNA seq demonstrate that DNA metabolic processes and cell cycle regulation are heavily enriched in the clusters (Figure S5). These data are not mentioned at all in the paper that focuses on more or less random changes in single genes. It would be tremendously beneficial to focus more on the broader changes than on the individual genes. Further, these alterations should be validated in other systems for example in vitro.

In the revised manuscript, we now focus upon alterations in pathways as opposed to changes in individual genes as we agree with this Reviewer that broader changes are likely more informative with regard to tissue biology. Additionally as noted above in relation to Reviewer 1 point #2, we now utilize an in vitro system to validate several pathways that are predicted to exhibit differential activity in basal and differentiated cells.

5. Analysis of clusters should be done in young cells only.

The primary goal of the current manuscript is to provide the first comprehensive interrogation of murine esophageal epithelium at the level of single cell resolution. To that end, we perform integrated clustering of all cells sequenced (regardless of age of the mice) and use these data to define the cell clusters present in murine esophageal epithelium (revised Figure 1) and to characterize basal, suprabasal, and superficial compartments overall (revised Figure 2). This approach allows for the inclusion of maximal data for these analyses. As only 130 genes display differential expression when comparing the transcriptional profile of young and aged murine esophageal epithelium, we further expect that the spectrum of cell populations present in aged and young mice will not differ.

6. All data should be made available as sortable excel sheets where it is possible to investigate the different proposed clusters.

We now provide these data as a Supplementary File Data S1.

7. S5 shows if pathways are upregulated or downregulated but S6 just shows if the pathways are significantly changed. S6 should show if the pathway is upregulated or downregulated. S6 shows the differences in each cluster while S5 does not. S5 should show the changes in each cluster individually.

We apologize for any confusion regarding the labeling of pathway analysis data in the initial submission. In the revised manuscript, we strictly utilize IPA for pathway analysis with color coding to indicate predicted activation state.

8. In several cases the most interesting results are not discussed. For example, "Gene ontology (GO) analysis of differentially expressed genes (DEG) in basal, suprabasal and superficial cell clusters further provided insight into the dynamic molecular signatures associated with lineage commitment in esophageal keratinocytes (Supplementary Figure S5A-D" and "Additionally, IPA provided insight into the pathways that are altered in the primary and minor branches (Figure 2E)". It would be of much greater interest to highlight these broader findings rather than the lengthy description of single gene changes.

In the revised manuscript, we now focus upon alterations in pathways as opposed to changes in individual genes. In addition to validating select pathways that were predicted by IPA to show differential regulation in basal cells compared to differentiated cells (revised Figure 2C, D), we now provide a discussion of the pathways that are identified by IPA to be differentially regulated in basal and superficial cells. We additionally clearly indicate the pathways predicted by IPA to be uniquely activated or inhibited in individual cell clusters (revised Figure 3B)

9. The organoid culture results are very surprising. It would be highly beneficial to be able to determine

experimentally if cells from young mice have a propensity to stay stem like while old cells have a propensity to undergo differentiation. It would be further interesting to quantify if there are more or less of the different clusters of cells with age in vivo in the squamous epithelium.

Reviewer 2 encouraged us to increase the sample size of our dataset. To do so, we doubled the number of both young and aged mice used for single cell RNA-sequencing in the revised manuscript such that the dataset now includes 44,679 epithelial cells (compared to 7,972 in the initial submission) across 10 mice (5 young, 5 aged). In this more robust dataset, we no longer see significant changes in basal cell populations when comparing young and aged mice. We have, however, included single cell sequencing of organoids in Figure 5 to determine the relationship of this experimental platform to our findings with regard to cell populations found in murine esophageal epithelium. We feel strongly that increasing the size of our dataset and including a comparative analysis of the cell populations in murine esophageal epithelium and organoids derived from this tissue has enhanced the impact of the revised manuscript.

10. For Figure 4 it is unclear why String analysis is chosen for looking at broader gene expression changes. It would be beneficial to stick with one analysis type (GO for example).

We now use IPA for all pathway analyses in the revised manuscript.

11. Altered mitochondria function is somewhat surprising given that esophageal changes are not normally seen in patients suffering from primary mitochondrial dysfunction. This should be discussed.

Emerging reports indicate that GI manifestations, including gastroesophageal reflux disease and achalasia, occur in patients with mitochondrial diseases, which we now noted in the revised discussion.

12. Many accounts of references to previous figures in the paper. This is quite confusing.

We apologize for any confusion that the structure of the initial submission caused. We have made substantial effort to improve the organization of the revised manuscript to make it more reader friendly.

Reviewer #2 (Remarks to the Author):

1. The esophageal epithelium of the mouse has been subject to intense study over the last decade. Here the authors apply single cell RNA sequencing to this tissue, seeking to establish the transcriptional states in the basal layer that contains proliferating cells and the upper layers that are undergoing differentiation. This is interesting as it builds on a wealth of transgenic modelling and the approach has proven informative in the similar stratified squamous epithelium of the oral cavity (PMID 30472156). The authors argue that there are multiple populations in the basal layer, new differentiation markers can be identified and that mitochondrial dysfunction is a feature of aging, a finding that is then followed up by functional studies.

We appreciate that the reviewer finds the study of interest and has the potential to add to our understanding of squamous epithelial biology across tissue types.

2. The data comes from 3 young and 3 aged mice, each of which was analyzed in technical duplicates. This gives the opportunity to look at the technical variation within each mouse, but it appears from Fig. S1G that the two runs were just summed. It would be helpful to present the intra-mouse sample variation.

In the revised manuscript we have increased our sample size to 5 young and 5 aged mice (from n=3/age group in the initial submission) with a total of 44,679 epithelial cells (compared to 7,972 in the initial submission) comprising our dataset. In the revised manuscript, we now show a comparison of the technical replicates that were run for each mouse in terms of transcript count, unique transcript count and percent mitochondrial genes (Supplementary Figure S2).

3. Initially, all the mice are then pooled for analysis. It is notable, however, that one mouse (aged 1) contributes little data while aged 2 has a far higher proportion of mitochondrial transcripts than the other animals. It is essential to know the extent to which the mitochondrial aging phenotype described here is

altered by the presence of this inter animal variation, especially given the limited contribution of mouse 1 to the data set. Given the variability of the data, adding 2-3 more aging mice may resolve this issue. Given these issues the validity of grouping all animals together needs to be carefully justified.

In the revised manuscript, we now include 5 mice per age group and show percent mitochondrial genes by replicate for each mouse (Supplementary Figure S2). We also performed sample integration at the mouse level - instead of the age group level - to discover a comparable and shared cell cluster structure that exists in all mice in the study.

4. Fig 2 A and B confirm that the basic structure of the tissue with basal, early and late differentiation is captured in the dataset. Immunostaining is used to confirm protein expression of selected markers on a basal to differentiated axis. Lineage reconstruction is performed in two principle components with separate analysis and clustering. Why were 2 PC selected? How robust are the clustering results to variation of the analysis parameters. Given that this is the bulk of the paper it is critical to investigate this, as the clustering may change.

In the revised manuscript we utilize Seurat and Monocle 3 for lineage reconstruction. Both use 30 PCs that are then used as input for UMAP. As UMAP is primarily a visualization tool, we chose a max of 2 dimensions (UMAP axes) for an easily understandable 2D trajectory. Notably, cells within our dataset cluster similarly in the UMAP object shown in revised Figure 1C, which was generated using Seurat with integrated clustering, and in the UMAP object shown in Figure 7B, which was generated using Monocle 3 with batch correction. Moreover, both show a trajectory from proliferative basal cells to post-mitotic terminally differentiated cells that is consistent with the known biology of squamous epithelial tissues.

5. The authors focus on 5 putative basal cell populations and their characteristics, exploring differences in GO terms. This is fraught with pitfalls as it depends on GO term assignments which is not necessarily reliable (https://doi.org/10.1007/978-1-4939-3743-1_14 and <https://www.nature.com/articles/s41598-018-23395-2>). For example, in Fig 2E are Col17a, Krt14, Krt15 and Krt5 really linked to cell cycle regulation and the activity of Plk or Krt4 and Krt14 to glutathione redox and the unfolded protein response? If so how? This analysis does not seem a robust way to infer biologically meaningful differences based on the genes shown and more critical assessment of this part of the study is required assuming it survives the tests of robustness above.

We fully appreciate the significance of investigating the functional relationships between identified markers of cell populations in murine esophageal epithelium and pathways predicted to be altered in these populations. In the revised manuscript, we use IPA to focus on evaluation of pathways, transcription factors, and kinases in the individual cell populations of esophageal epithelium as well as in the basal, suprabasal, and superficial cell compartment collectively. We further validate select pathways predicted to display differential activation in basal and superficial cells. We believe strongly that this study represents a significant advance in the field and that the described analyses of transcriptomic data will be hypothesis-generating not only for our own lab but for the broader esophageal biology community. Indeed, we are actively working toward executing functional investigations with regard to the roles of (1) identified subsets of cells (with basal cells being of particular interest) under conditions of homeostasis as well as in the context of esophageal disease models; (2) pathways predicted to be differentially regulated during squamous cell differentiation (e.g. EIF2, Glutathione-mediated detoxification). The discussion in the revised manuscript now includes an acknowledgment of the limitations of the current study with regard to inference of functional roles for pathways identified to be differentially regulated across the dataset.

6. In Fig 3 GO terms are used to argue for the existence of a quiescent basal cell population. This is odd, as transgenic proliferation assays reveal no evidence of such a population in mouse esophagus (PMC3527005, PMC7080751). Given this data, how reliable is the assignment of quiescence based on GO analysis? It is surely essential to constrain analysis of high dimensional data using experimental data where this is available. This section should be re-evaluated in the light of the literature on the esophageal epithelium.

We very much appreciate that Reviewer 2 challenged our prior claim that data from the current study supports the existence of a quiescent cell population. After careful review of the literature and our own data set, we agree that this claim was an overinterpretation of our data. The revised manuscript no longer includes this claim.

7. Figure 4 compares young and aged animals. If this analysis is to be performed more mice should be analysed as discussed above. The attribution of mitochondrial differences may reflect the marked variation in % mitochondrial transcripts across animals.

As discussed with regard to Reviewer 2 point 2, we have increased the sample size of young and aged mice from 3 per age group to 5 per age group. Additionally, we now show percent mitochondrial expression data for each individual replicate across all animals in Supplementary Figure S2.

8. The organoid data may be interesting, but needs more characterisation. The statement that ‘altered basal cell dynamics...(are) a feature of esophageal aging’ would not seem to be reliable given this rests on second passage organoid cultures.

After increasing our sample size in the revised manuscript, we no longer detect any significant changes in the representation of any basal cell cluster when comparing young and aged mice. We do, however, perform single cell RNA-Seq of murine esophageal organoids and compare the data obtained in this experimental platform to data obtained in murine esophageal epithelium (Figure 5). This comparison indicates that 10 of the 11 cell types identified in murine esophageal epithelium are recapitulated in 3D organoids with varying degree of transcriptional similarity. Additionally, the representations of several clusters is different when comparing murine esophageal epithelium to 3D murine esophageal organoids. These data support 3D organoids as a viable model for the study of esophageal biology, particularly with regard to basal cells.

9. The discussion is overlong and does not seek to compare the analysis here with similar epithelia in the mouse (oral and interfollicular epidermis) where functional and scRNAseq data are available.

In the revised manuscript, we refined the discussion which now includes a section comparing our dataset with published findings in murine oral and follicular epidermis.

10. Overall while a comprehensive scRNA seq analysis of mouse esophagus with aging would be a useful resource, this cannot be built on such a small number of animals with variable scRNAseq quality.

With the increased sample sizes for young and aged mice as well as incorporation of the Reviewer’s comments regarding data analysis, interpretation, validation, and discussion of relationship to existing literature we now provide a robust and comprehensive single cell RNA-Seq analysis of murine esophageal epithelium.

Reviewer #3 (Remarks to the Author):

In this paper, Kabir et al perform ssRNA sequencing of the homeostatic murine esophagus of young and aged mice. They have identified several epithelial subpopulations with distinct molecular signatures, revealed an epithelial differentiation program, and observed disbalance in epithelial differentiation particularly of the basal cells as well as impaired mitochondrial activity in aged mice. The paper provides a detailed perspective on the epithelial heterogeneity in the murine homeostatic esophagus and advances ssRNA sequencing as an important tool to further explore esophageal biology.

We thank this reviewer for their kind words.

Comments:

1. The authors utilize esophageal organoids from young and aged mice to demonstrate disbalance in the basal cell subpopulations observed in the aged mice. However, the relevance of this system to serve as a functional

readout for cellular heterogeneity in the esophageal epithelium is unclear. Which cells form the organoids? It is reasonable to assume that initially, the epithelial cells need to proliferate to form the organoid. Does this suggest that quiescent basal cluster 5 is lost? What happens during differentiation? Is heterogeneity of the epithelial cells, in particular basal cells, observed in the fully differentiated organoids? As the authors show, the proliferation rate of the organoids from the aged mice is increased with passages, which is inconsistent with the increased amount of quiescent cells from the basal layer 5 in the aged mice. To answer these questions scRNAseq of the organoids should be performed and compared to the signatures of the epithelial populations in the esophagus.

After increasing our sample size in the revised manuscript, we no longer detect any significant changes in the representation of any cell cluster population when comparing young and aged mice. We do, however, perform single cell RNA-Seq of murine esophageal organoids and compare the data obtained in this experimental platform to data obtained in murine esophageal epithelium (Figure 5). This comparison indicates that 10 of the 11 cell types identified in murine esophageal epithelium are recapitulated in 3D organoids with varying degree of transcriptional similarity. Notably, while all basal cell populations are predicted to be present in 3D organoids, the representation of 2 basal clusters is decreased in organoids while that of an additional basal cluster is increased. These data support 3D organoids as a viable model for studying esophageal biology as noted in the revised discussion.

2. It is important to address fundamental differences in the esophageal epithelial biology between humans and mice in the introduction and discussion of the paper. This is especially relevant to the properties of the basal layer cells in contact with lamina propria that remain mostly quiescent in humans. The relevance of the results of the study to human vs mouse esophageal epithelial biology should be highlighted.

We now evaluate the representation of cell populations identified by the current study in murine esophageal epithelium with a dataset from human esophageal epithelium (Figure 6). We further discuss the implications for these data in the revised discussion.

3. In several places in the paper, the description of the results is vague. For example, p7 line 6 "...provided insight into the dynamic molecular signatures associated with lineage commitment.." This statement refers the reader to the figure that is overloaded with data and hard to follow. Please state clearly the main findings of the analysis and label the figure accordingly. Another example is p7 line 11 related to the individual molecular identity of 8 clusters. Please detail these findings and clarify Figure 2C accordingly. Another example, page 8 line 6 "...provided insight.." What is the finding here?

In the revised manuscript, we have made substantial effort to improve the clarity of the text and to more thoroughly connect the text to the data presented.

4. Related to above point, it remains unclear which molecular markers separate basal subpopulations. Figure 2C does not seem to be clustered in any way and is hard to follow. Please simplify or color code relevant pathways.

We appreciate that original Figure 2D, which demonstrates pathways analysis on DEGs in the individual epithelial clusters identified in murine esophageal epithelium, is quite busy. To more clearly indicate pathways that are predicted to be uniquely activated or inhibited in each individual cell cluster, we now provide a summary figure (revised Figure 3B).

5. Similarly, it is unclear how many genes were differentially expressed between aged and young mice. Was it 64 genes as stated on p11 line 15?

In the revised manuscript we have substantially increased the number of cells sequenced in each mouse as well as the number of animals used for sequencing. Analysis of the resulting data revealed 130 differentially genes when comparing young and aged animals. This is noted on Page 9, Line 5.

6. The authors use published ssRNA seq data for the human esophagus to assess the expression of the basal, suprabasal, and superficial markers identified in the paper. They conclude that these markers “display appropriate localization” in the human epithelium. I disagree with the conclusion. Based on the annotation of the data, expression of COL17A1 is not primarily basal and CNFN seems to be expressed throughout the esophageal epithelium. I am not sure how this data set confirms the specificity of the expression at least for these markers. Related to it, Col17a1 has previously been shown as a basal marker for skin and esophageal epithelial cells (PMID: 31451683; PMID: 32187560). The differences between mouse and human also need to be considered.

As described in reference to Reviewer 3 point #2, the revised manuscript includes evaluation of our murine esophageal epithelial dataset in relation to scRNA-Seq data from human esophageal biopsy specimens (Figure 6) as well as a discussion of these findings. Additionally, in light of this Reviewer’s concern about the localization of COL17A1 and CNFN in the human dataset we have further opted to remove the data evaluating expression of these genes in the human dataset. Moreover, while COL17A1 has previously been shown as a marker of basal cells in human esophagus, it has not (to the best of our knowledge) been demonstrated to mark murine basal cells in the esophagus, which is important given the differences between human and mouse as noted by this Reviewer. We further discuss this in the context of comparison of murine and human esophageal epithelium in the revised manuscript.

7. Please clarify the significance of minor vs major branches identified in the pseudotime analysis.

In the revised manuscript, we now employ Monocle 3 for lineage reconstruction by pseudotime (Figure 7). Using Monocle 3-based pseudotemporal trajectory inference, cluster identification, and evaluation of cell cycle-associated genes, we now provide a working model of cell fate determination in the esophagus (Figure 7D). We propose that the pseudotime trajectory initiates in the G1/G0-enriched basal population 6, before moving to basal population 1 which is enriched for S phase-associated genes, then continuing through basal populations 2 and 3, both of which are enriched for G2/M-associated genes. As cells exit the cell cycle they then return to basal population 6, which represents a decision point at which cells may (1) re-enter the cell cycle, (2) execute squamous cell differentiation; or (3) remain in G0/G1 phase. These are classified as the 3 branches of the pseudotime in the revised manuscript with no minor branches.

8. Please label epithelial clusters in figure 3E.

Figure 3E in the initial submission depicted cell cycle genes across the Seurat UMAP. This figure was shown in the context of a pseudotime projection of the basal cell clusters only. We realize that this made integration of the two pieces of data difficult. In the revised manuscript, we now show a Monocle 3-based UMAP object depicting the 11 cell clusters in esophageal epithelium (Figure 7B) alongside the corresponding pseudotime projection (Figure 7A) and cell cycle gene projection (Figure 7C). As the cluster assignments using either Seurat or Monocle 3 are not always discrete (e.g. in revised Figure 7B the basal 5 population is dispersed among basal 6 cells) it is not possible to clearly demarcate the individual cell clusters on the cell cycle gene projection. We do, however, believe that inclusion of the cluster UMAP and cell cycle progression in the same figure as described will allow the reader to better integrate these two pieces of data.

9. Please explain the impact of peeling the skin layer; how specific is this to the epithelium and how sensitive to the whole epithelium? Histological imaging before and after peeling, as well as assessment of the remaining non-peeled tissue for bulk RNAseq would assess this.

In the revised manuscript, Supplementary Figure S1 provides histology and bulk RNA-Seq data on peeled epithelium-enriched mucosa layer and muscle layer and underlying muscle layers. The data suggest that the epithelium-enriched fraction contains the whole epithelium as well as fibroblasts while the muscle layer appears to be exclusively composed of muscle.

10. In figure 1B, Why are the authors only averaging expression only for each cluster? Single-cell-wise expression heatmap would be informative.

In the revised manuscript, we now include a single-cell expression heatmap (Supplementary Figure S4).

11. In supplementary Figure 2, Mouse 1/2/3 seems to be biological replicates. Then what is "Replicate 1 and 2"? Figure legends says "2 reads were done for each mouse". Are they sequencing replicate or separate library preparation?

In the initial submission replicates were technical sequencing replicates performed for each mouse. In the revised manuscript, we now address reproducibility both in the pre-processing steps as well as during post-hoc analyses. Specifically, we now integrate the dataset at the sample level to find a comparable biological set of cell populations that exist across all of the mice in our dataset.

12. In figure 1, reproducibility across biological replicates are not addressed except for the Fig S1F. More quantitative assessment is needed. for example cell % in each cluster per sample. This is critical information because there can be a cell cluster originated from biological variability.

We now show each mouse as a dot in the scatter plots depicting cluster proportion in young and aged mice (Figure 4B).

13. Color-scheme is confusing (especially basal 2 vs 5). Direct annotation on top of the UMAP is strongly recommended.

In the revised manuscript we have altered the color scheme to allow for improved visualization of the individual clusters in UMAP objects. Additionally, in revised Figure 3B, we provide a summary of unique pathways for each cluster in association with the UMAP plot.

REVIEWERS' COMMENTS

Reviewer #1 (Remarks to the Author):

I thank the authors for answering my concerns. I think their efforts have substantially improved the manuscript. A few minor comments:

-Of all the age-associated changes only one pathway shows alterations in mitochondrial function(oxidative phosphorylation). Perhaps one could consider whether mitochondrial function contributes in any meaningful way to the aging epithelium. Stress signaling (ferroptosis, nrf-2 mediated stress response, IL-8 signaling, glutathione-mediated detox.) seem to be much more enriched than this sole pathway. Perhaps the title of the paper should be changed?

-The authors have now included reflux and achalasia as examples of mitochondrial dysfunction leading to epithelium dysfunction. Both conditions are not caused by mitochondrial changes in the epithelium but rather mitochondrial dysfunction in tissues surrounding the epithelium (muscle and nerves). Reflux and achalasia therefore do not support their claim of a mitochondrial involvement in epithelial dysfunction at all and should be removed.

Reviewer #2 (Remarks to the Author):

The authors have substantially improved the manuscript. By expanding the number of animals and simplifying the analysis the paper is much easier to follow and the findings are more robust. The moderating of the claims about mitochondrial ageing and stem cells are appropriate in the light of the literature. The data is fully accessible and will be a useful resource for others studying the esophagus and other tissues in the mouse.

A couple of minor points:

It would be worth citing McGinn et al Nature Cell Biol 2021 (PMID: 33972733) which includes scRNA seq data on developing and adult mouse esophagus.

The discussion on redox and somatic mutations page 16 lines 14-19 is hard to follow. If redox was implicated in human somatic mutations this would surely be manifest as the mutational signature SBS18, whereas the predominant signatures are SBS1 and 5? This should be rewritten or left out.

Reviewer #3 (Remarks to the Author):

Thanks for the thorough response

** See Nature Portfolio's author and referees' website at www.nature.com/authors for information about policies, services and author benefits

RE: NCOMMS-21-04499B; ‘Single cell transcriptomic analysis reveals cellular diversity of murine esophageal epithelium and age-associated mitochondrial dysfunction’ by Kabir MF and Karami A, et al.

REVIEWER COMMENTS

Reviewer #1 (Remarks to the Author):

I thank the authors for answering my concerns.

I think their efforts have substantially improved the manuscript. A few minor comments:

-Of all the age-associated changes only one pathway shows alterations in mitochondrial function(oxidative phosphorylation). Perhaps one could consider whether mitochondrial function contributes in any meaningful way to the aging epithelium. Stress signaling (ferroptosis, nrf-2 mediated stress response, IL-8 signaling, glutathione-mediated detox.) seem to be much more enriched than this sole pathway. Perhaps the title of the paper should be changed?

We have changed the title of the manuscript to ‘Single cell transcriptomic analysis reveals cellular diversity of murine esophageal epithelium.’ We have also added a sentence to the discussion noting the enrichment of stress response pathways in esophageal epithelium of aged mice.

-The authors have now included reflux and achalasia as examples of mitochondrial dysfunction leading to epithelium dysfunction. Both conditions are not caused by mitochondrial changes in the epithelium but rather mitochondrial dysfunction in tissues surrounding the epithelium (muscle and nerves). Reflux and achalasia therefore do not support their claim of a mitochondrial involvement in epithelial dysfunction at all and should be removed.

We have removed these claims.

Reviewer #2 (Remarks to the Author):

The authors have substantially improved the manuscript. By expanding the number of animals and simplifying the analysis the paper is much easier to follow and the findings are more robust. The moderating of the claims about mitochondrial ageing and stem cells are appropriate in the light of the literature. The data is fully accessible and will be a useful resource for others studying the esophagus and other tissues in the mouse.

A couple of minor points:

It would be worth citing McGinn et al Nature Cell Biol 2021 (PMID: 33972733) which includes scRNA seq data on developing and adult mouse esophagus.

We have added this citation.

The discussion on redox and somatic mutations page 16 lines 14-19 is hard to follow. If redox was implicated in human somatic mutations this would surely be manifest as the mutational signature SBS18, whereas the predominant signatures are SBS1 and 5? This should be rewritten or left out.

We have removed this portion of the discussion as it is speculative.

Reviewer #3 (Remarks to the Author):

Thanks for the thorough response.

We sincerely thank the three Reviewer’s for their thoughtful consideration of our manuscript which is now substantially improved.